

# From Monsoon to marine productivity in the Arabian Sea : insights from glacial and interglacial climates

Priscilla Le Mézo[1], Luc Beaufort[2], Laurent Bopp[1], Pascale Braconnot[1], and Masa Kageyama[1]

[1]LSCE/IPSL, UMR 8112 - CEA/CNRS/UVSQ, Centre CEA-Saclay, Orme des Merisiers, 91191 Gif-sur-Yvette, France
[2]CEREGE, UMR 7330, CNRS-IRD-Aix Marseille Université, Av. Louis Philibert, BP80, 13545 Aix en Provence, France
*Correspondence to:* Priscilla Le Mézo (priscilla.le-mezo@lsce.ipsl.fr)

**Abstract.** The Indian monsoon is known to boost biological productivity in the Arabian Sea. This paradigm has been extensively used to reconstruct past monsoon variability from paleo-proxies indicative of changes in surface productivity. Here, we test this paradigm by simulating changes in marine primary productivity for 8 contrasted climates from the last glacial-interglacial cycle. We show that there is no straightforward correlation between boreal summer productivity of the western and

central Arabian Sea and summer monsoon strength across the different simulated climates. Locally, productivity is fueled by nutrient supply driven by Ekman dynamics depending on both wind stress and wind stress curl. In our simulations, a stronger monsoon with intensified wind stress on the Arabian Sea can lead to either increased or reduced productivity depending on the exact ocean surface on which increased wind stress and a positive wind stress curl are acting. The effects of winds onto the ocean are modulated by the Indian summer monsoon intensity and pattern (e.g. position of the low-level jet over the Arabian

Sea), which in turn are driven by the orbital parameters and the ice sheet cover. The orbital parameters are indeed shown to impact wind stress intensity in the Arabian Sea through large scale changes in the meridional gradient of upper tropospheric temperature. But both the orbital parameters and the ice sheets affects the pattern of wind stress curl through the position of the sea level depression barycentre over the monsoon region (20°W-150°W, 30°S-60°N). The combined changes in monsoon intensity and pattern lead to higher glacial productivity during the summer season, in agreement with some paleo-productivity

reconstructions.

## 1   Introduction

The Arabian Sea biological productivity is influenced by the strong seasonal activity of the atmospheric circulation (Schott and McCreary, 2001; Ivanova et al., 2003; Lee et al., 2000; Luther et al., 1990). During the boreal summer, the Southwest monsoon consists of strong winds blowing from the south-west to the north-east of the Indian Ocean. These winds result from

the rapid heating of the landmass relative to the ocean, which creates a pressure gradient between the southern Indian Ocean high pressure cell and the low pressure cell over the Tibetan Plateau. During this season, heavy precipitation occurs in India and south-east Asia. In the Arabian Sea, the alongshore winds off the coast of Somalia focus into a low-level jet, called the Somali Jet and generate a strong coastal upwelling (Anderson et al., 1992; Findlater, 1969). In addition, the positive curl of the wind stress between the axis of the jet and the western coast induces Ekman pumping (Murtugudde et al., 2007; Barber et al.,



2001; Anderson et al., 1992; Findlater, 1969). These two processes are responsible for increased productivity in the western coastal Arabian Sea due to a higher supply of nutrients to the surface layer (Anderson et al., 1992; Anderson and Prell, 1992). The upwelled nutrients are advected from the coast to the north and the interior of the sea. So, productivity in the central and northern Arabian Sea also increases during the Southwest Monsoon (Caley et al., 2011; Prasanna Kumar et al., 2001; Keen

et al., 1997). Wind stress and mixing of the upper layers, as well as Ekman pumping generated by the positive wind stress curl, also contribute to the supply of nutrients to the surface layers and increase productivity in those regions (Resplandy et al., 2011; Wiggert et al., 2005; Prasanna Kumar et al., 2001; Lee et al., 2000).

Monsoon intensity can be characterised in different ways, depending on the observational scale and on the studied processes. Precipitation is a major indicator of monsoonal changes. For example, the rainfall-based index defined as the seasonally av-

eraged precipitation over all the Indian subcontinent from July to September, is used to monitor the strength of the monsoon over India (Mooley and Parthasarathy, 1984). A second indicator of the monsoon strength is based on the sea level pressure (SLP) that is a large-scale fingerprint of the monsoon. The monsoon strength can be determined by the SLP anomaly gradient between a northern region over the Tibetan Plateau, where the Tibetan Low develops during the monsoon months, and a southern region over the southern Indian Ocean, where the Mascarene high develops. The large-scale changes of SLP impact the

local dynamics over the Arabian Sea (Schott and McCreary, 2001). Monsoon intensity can also be related to the strength of the winds over the Arabian Sea and the associated upwelling. The general paradigm is that a stronger summer monsoon generates stronger upwelling that enhances productivity. Based on this paradigm, past monsoon intensities have been reconstructed using proxies of productivity from marine sediment cores (Caley et al., 2011; Ivanova et al., 2003; Clemens and Prell, 2003).

Monsoon reconstructions and modelling studies (Marzin and Braconnot, 2009; Braconnot et al., 2008; Anderson and Prell,

1992; Prell et al., 1992) have shown that insolation variations are the major driver of fluctuations in the summer monsoon intensity : the monsoon is stronger when the Northern Hemisphere summer insolation is higher (e.g. during the Holocene). Changes in the orbital parameters, such as the precession that is defined as the longitude of the perihelion, or the obliquity that is defined as the angle between the equator and the orbital plane, modify the seasonal cycle of insolation. Along with orbital parameters, changes in ice-sheet height also have an impact on the monsoon intensity (Masson et al., 2000; Emeis et al., 1995;

Prell et al., 1992; Anderson et al., 1992).

There has been some concern about the fact that marine proxies for productivity may be influenced by other processes than monsoon intensity, such as changes in ice volume, aeolian transport of nutrients or the Atlantic Meridional Overturning Circulation (Ruddiman, 2006; Ziegler et al., 2010; Caley et al., 2011). Moreover, most studies linking monsoon and productivity in the past have focused on the monsoon intensity but, the monsoon pattern, e.g wind orientation, can also change in time.

Sirocko et al. (1991) have shown that summer monsoon mean position was shifted southward during glacial periods. The monsoon pattern affects the position and the orientation of the low-level jet over the Arabian Sea, which modifies the upwelling of nutrients in the Arabian Sea (Anderson and Prell, 1992). Furthermore, Bassinot et al. (2011) showed opposite evolutions of the upwelling behaviour in the western coastal Arabian Sea and the south-western tip of India during the Holocene, which they related to a southward shift of the monsoonal winds.



Here, we investigate the relationship between the summer monsoon intensity and the Arabian Sea biological productivity. How do changes in the summer monsoon structure along with changes in its intensity impact productivity in the Arabian Sea ? Could the variations in the summer monsoon structure explain higher productivity rates in some glacial climates ? To answer these questions, we test the effects of a range of astronomical parameters and different ice-sheet states on the Arabian Sea

productivity.

In section 2, we describe the model we use and the experiments we performed, we evaluate the model results for the pre-industrial and detail the analyses we performed. In section 3, we explain the changes in productivity in the Early Holocene and then, look at several glacial and interglacial climates to link productivity changes to local dynamics and boundary conditions. In section 4, we discuss our results in the light of the summer monsoon paradigm and, we perform a simple model-data comparison

and discuss the effects of seasonality on productivity. Finally we summarise our results and give some perspectives.

## 2  Model, experiments, evaluation and diagnostics

### 2.1  The model

This study uses an Earth System Model (ESM) that explicitly represents the global climate, oceanic circulation and marine productivity. We use the IPSL-CM5A-LR model developed at the Institut Pierre Simon Laplace (IPSL) (Dufresne et al., 2013).

This ESM is composed of the atmospheric general circulation model LMDZ5A (Hourdin et al., 2013) coupled to the land-surface model ORCHIDEE (Krinner et al., 2005) and the ocean model NEMO v3.2 (Madec, 2011), which includes the ocean general circulation model OPA9, the sea-ice component LIM-2 (Fichefet and Maqueda, 1997) and the biogeochemical model PISCES (Aumont and Bopp, 2006). These components are coupled once a day using the OASIS coupler (Valcke, 2012).

We use the low-resolution (LR) version of the model with a regular atmospheric grid of 96×96 points horizontally and 39

vertical levels and, an irregular horizontal oceanic grid (ORCA2.0) with 182×149 points corresponding to a nominal resolution of 2°, enhanced near the equator and over the Arctic and sub-polar North Atlantic. The ocean vertical grid comprises 31 levels with intervals from 10 meters for the first 150 meters and up to 500 meters for the bottom of the ocean.

The PISCES model simulates marine bio-geochemistry and lower trophic levels. PISCES includes two phytoplankton types (nano-phytoplankton and diatoms), two zooplankton size-classes (micro- and meso-zooplankton) and two detritus compart-

ments distinguished by their vertical sinking speed (small and large organic matter particles), a dissolved organic carbon pool, and five nutrients (Fe, $NO_3$, $NH_4$, Si, and $PO_4^{3-}$) (Aumont and Bopp, 2006). In PISCES, phytoplankton growth is a function of temperature, light, mixed layer depth and nutrients.

### 2.2  Experiments

Here, we exploit 8 simulations of IPSL-CM5A-LR forced by different boundary conditions (orbital parameters, gas concentra-

tions and ice sheets cover), to account for different climates throughout the last glacial-interglacial cycle, as detailed in Table 1 and Figure 1.



The reference simulation (CTRL), is a pre-industrial climate with no external forcing such as volcanoes or anthropogenic activities (Dufresne et al., 2013), forced by pre-industrial CMIP5 forcings (Taylor et al., 2012) and the present day ice sheet (0k on Figure 1). A mid-Holocene (MH) simulation, 6 kyr BP (Kageyama et al., 2013), part of Paleoclimate Modeling Inter-comparison Project phase 3 (PMIP3) (Braconnot et al., 2012) and an early-Holocene (EH) simulation, 9.5 kyr BP, are used

to study productivity changes in different interglacial climates. The EH simulation trace gas concentrations are the same as for the CTRL simulation whereas $CH_4$ and $N_2O$ concentrations are slightly lower for the MH simulation compared to CTRL (Table 1). MH and EH simulations mainly differ in their astronomical parameters, especially in the precession value (Table1, Fig. 1). Both Holocene simulations are forced by the present-day ice sheet cover (Fig. 1). Four glacial simulations have also been performed for; the last glacial maximum (LGM, 21 kyr BP), the isotopic stage 3 (MIS3, 46 kyr BP) and three isotopic

stage 4 states : MIS4F (60 kyr BP), MIS4M (66 kyr BP) and MIS4D (72 kyr BP). The LGM, which has also been performed for PMIP3 (Kageyama et al., 2013), has the largest ice sheet (Fig. 1) (Abe-Ouchi et al., 2015), which modifies the land-sea distribution and topography since the sea-level is reduced by about 120 meters. The LGM run has the lowest greenhouse gas concentrations of this set of 8 simulations (Table 1). Two of the three MIS4 simulations (MIS4F and MIS4D) are described in Woillez et al. (2014). The MIS4 ice-sheets have been prescribed by using the 16 kyr BP ice-sheet, which is the period for which

we have an ice sheet reconstruction for the same sea level as during MIS4, i.e. 70 meters lower than today (Ice6g-16k on Fig. 1) (Peltier et al., 2015). This is the most realistic we could do given the available reconstruction at the time of running the MIS4 experiments (Woillez et al., 2014). However, our MIS4 runs are different from the ones described in Woillez et al. (2014) since we added the nutrient inputs from dust, rivers and sediments that are essential to marine productivity. Large changes in preces-sion occur between the three MIS4 simulations (Table 1, Fig. 1). The MIS3 simulation uses the same ice-sheet reconstruction

as MIS4 and it has the lowest eccentricity and highest obliquity of all 8 simulations (Table 1, Fig. 1).

In PISCES, three source terms contribute to the input of nutrients in the ocean: atmospheric dust deposition, river input and sediment mobilisation. The change in sea-level in glacial climate simulations modifies the land-sea mask, thus in the LGM, MIS3 and all MIS4 (F,M,D) simulations, the source terms were adjusted so that the ocean receives the same quantity of associated nutrient supply as in CTRL. In these simulations, no attempt was made to account for the dustier glacial states

(Bopp et al., 2003). All productivity changes are therefore due to other factors.

Our analyses are performed on 100 years of monthly outputs from the last stable part of each simulation.

### 2.3   Modern evaluation of the summer mean

This study focuses on primary productivity in the Indian Ocean for the last glacial-interglacial cycle as simulated by the IPSL-CM5A-LR coupled model. Figure 2 shows the seasonal cycle of productivity for three areas in the Arabian Sea: a coastal area

in the western Arabian Sea (Fig.2a, orange area), an area that covers the central-east Arabian Sea (Fig.2a, grey area) and a northern region (60°E-68°E, 20°N-68°N). Productivity has two periods of bloom: one in summer and one in winter. In both the coastal and central regions, the summer season is the most productive period of the year (Fig.2b) and contributes the most to the sediments bulk composition. In the northern Arabian Sea, both seasons are equally productive (Fig.2b). In boreal winter, the mechanisms behind productivity changes are different compared to summer ones. The winds reverse and nutrients are brought





to the surface thanks to vertical mixing and advected from the central sea to the coast. During this period, productivity is high in the north-western Arabian Sea (Fig.2b). Following these observations, we focus our analyses on the boreal summer season, defined as June-July-August-September (JJAS) to account for the whole summer monsoon, and we will especially analyse the coastal and central areas (orange and grey shadings on fig.2a).

A comparative global evaluation of the marine bio-geochemical component of the ESM has been published in Séférian et al. (2013). Even if the model poorly represents the deep-ocean circulation, especially in the Southern Ocean, it has a quite good representation of annual wind patterns, wind stress, mixed-layer depth and geostrophic circulation. The model is able to represent the global ocean biological fields such as macro-nutrients, with correlations higher than 0.9, and surface chlorophyll concentration, with a correlation coefficient of 0.42 (Séférian et al., 2013). We focus here on the representation of the physical

processes and productivity distributions in the Indian Ocean, especially in the Arabian Sea. We use satellite products from remote sensing by NASA?s Sea-viewing Field-of-view Sensor (SeaWIFS) during the period 1998-2005 for monthly productivity (Lévy et al., 2007) and the NOAA Multiple-Satellite 1995-2005 climatological cycle for wind intensity and wind stress (Zhang, 2006). We compute the observed and modelled wind stress curl intensity from the wind stress data and model output, respectively. We compare the observations to the pre-industrial (CTRL) simulation outputs.

Figure 3a shows that simulated boreal summer productivity integrated over the whole water column is underestimated relative to the reconstructed boreal summer productivity, especially in the regions of upwelling, along the coast of the Arabic Peninsula and Somalia. The spatial correlation coefficient, $R$, between the observed and simulated productivity is 0.44. Underestimation of productivity is first caused by an underestimated wind intensity (Fig. 3b), which affects the extent and intensity of the coastal upwelling and the supply of nutrients to the surface layer. The boreal summer wind patterns, which are char-

acteristic of the boreal summer monsoon system, are better represented than productivity with a correlation of 0.86. Sperber et al. (2013) studied the representation of the Asian summer monsoon in the CMIP5 models, which comprises the IPSL model. They showed that the monsoon was better represented in the CMIP5 models compared to the CMIP3 models, especially the monsoonal winds. We can however note that the alongshore winds in the western Arabian Sea have a more northerly orientation in the CTRL simulation than in the observations, which can affect the dynamical processes in the region (Fig. 3b).

In the Arabian Sea, summer productivity is affected by the winds through different mechanisms implying the wind stress and the wind stress curl (Anderson et al., 1992). The strong winds along the Arabian coast, called the Somali Jet, generate a positive wind stress which increases Ekman transport off the coast. The water that leaves the coastal area is being replaced by subsurface water: this is the coastal upwelling. Similarly to the wind intensity, the CTRL simulation wind stress intensity is underestimated compared to the reconstructions : the maximum wind stress intensity is lower and it does not extend far north in

the Arabian Sea as the reconstructed wind stress (Fig.3c). The wind stress orientation is also more zonal in the simulation than in the observations (Fig. 3b,c). Figure 3d represents the wind stress curl, computed from the wind stress, in the simulation and in the observations. The simulated distribution resembles the reconstructed one: on the left hand side of the strong low-level wind jet, between the coast and the maximum wind intensity, the curl in the wind stress is positive and, on the other side of the jet the wind stress curl is negative. The differences seen in the jet position and width are transmitted to the wind stress and

wind stress curl intensity and distribution.





Discrepancies between our pre-industrial simulation and the observations may be attributed to the model coarse resolution. In Resplandy et al. (2011), a higher resolution version of the model was used to study the effects of meso-scale dynamics on productivity. They showed that the model is able to reproduce the observed meso-scale dynamics, such as the Great Whirl and filaments that transport nutrients from the coast to the open sea. They highlighted the major role of the eddy-driven transports

in the establishment of biological blooms in the Arabian Sea and the model ability to represent the different physical processes at stake behind productivity blooms in summer and in winter in the region. Nevertheless, even though both the winds and productivity are underestimated in CTRL by the lower resolution version of the model, the physical mechanisms playing a role in the marine productivity are represented, which is therefore adapted to our study.

## 2.4 Diagnostics

In this section we briefly describe the variables and the methods we use throughout the paper. We are interested in the pathways between the large-scale Indian summer monsoon system and the Arabian Sea primary productivity. To characterise the boreal summer monsoon intensity, we use the meridional gradient of upper tropospheric temperature (TT, averaged from 200 to 500 hPa) between a northern region covering India, south-east Asia and the Tibetan Plateau (60°E-120°E, 10°N-45°N) and a southern region over the tropical Indian Ocean (60°E-120°E, 25°S-10°N) (Marzin and Braconnot, 2009; Goswami et al., 2006).

This gradient, $\Delta$TT, is associated with the temperature land-sea contrast (Marzin and Braconnot, 2009). $\Delta$TT is averaged over the boreal summer period (JJAS for June-July-August-September) and the higher its value, the stronger the Indian summer monsoon.

Changes in the monsoon intensity and pattern affect the sea level pressure (SLP) field. We compute the SLP from the model outputs (i.e. model air temperature and pressure, within and between the atmospheric grid levels, and orography) using the

extrapolation described in Yessad (2016). We define "SLP anomalies" as the SLP minus the global annual average of SLP. In order to characterise the monsoon pattern, we compute the barycentre of the region defined by an SLP anomaly lower than -5 hPa over the region covering the African, East Asian and Indian monsoons regions of influence (20°W-150°E, 30°S-60°N) and we call "SLPa-5" the region delimited by the -5 hPa contour in SLP anomalies. This SLPa-5 barycentre is representative of the balance between the different monsoons as well as of the Indian summer monsoon wind position and direction over the

Arabian Sea. A modification of the monsoon pattern, which can have impacts on productivity through the atmospheric forcing onto the ocean circulation, can then be related to movements of the SLPa-5 barycentre. We only focus on the Tibetan Low since the Mascarene High, the region of high SLP in the southern Indian Ocean, barycentre remains quite similar in the different simulations.

Anderson et al. (1992) showed that the wind stress intensity generates coastal upwelling and that the positive wind stress curl

is responsible for Ekman pumping offshore. In the central Arabian Sea, the negative wind stress curl generates downwelling. We focus our work on these two wind variables and on two areas in the Arabian Sea: the coastal area in the western Arabian Sea, which covers the region of positive curl, for the CTRL climate, between the axis of the Jet and the coast (Fig.2a, orange area) and the area covering the central-east Arabian Sea (Fig.2a, grey area).



In the following sections of the paper, total primary productivity (TPP) is defined as the sum of nano-phytoplankton and diatoms productivity integrated over the whole water column. We also analyse nitrate concentrations in the first 30 meters of the water column, nitrate being the major limiting nutrients in the region and, its supply to the surface layers being mainly driven by atmospheric changes via coastal upwelling and Ekman pumping.

We use the CTRL simulation as a reference. All changes are then defined relative to this pre-industrial simulation.

## 3   Simulated paleoproductivity and monsoon changes

In this section, we investigate the changes in summer productivity in past climate simulations with respect to the CTRL simulation, starting with the early Holocene and then generalising to all the climates.

### 3.1   The early Holocene case

The early Holocene (EH) experiences a stronger Indian summer monsoon than the pre-industrial (Fig.4) therefore, we would expect higher productivity in the Arabian Sea. However, the EH simulation shows lower levels of productivity than in CTRL (Fig.5). We explain this counter-intuitive result by a change in the monsoon pattern instead of a change in its intensity (Fig.6).

The early Holocene, which we choose to represent with a snapshot at 9.5 kyr BP, is an interglacial period that mainly differs
from the pre-industrial because of the imposed obliquity (24.2306° vs 22.391°) and precession (303.032° vs 102.7°) (Table 1). These changes in astronomical parameters cause the boreal summer insolation in the northern hemisphere to be higher than in the pre-industrial climate (Marzin and Braconnot, 2009). In EH, the boreal summer (JJAS) northern hemisphere (0-90°N, NH) mean insolation is 20 $W.m^{-2}$ higher than in CTRL (Fig. 4a,b). This change in insolation modifies the upper tropospheric temperature gradient, $\Delta TT$, represented on Figure 4c,d. In EH, $\Delta TT$ is 1 K higher than in CTRL supporting a stronger monsoon
intensity consistent with previous studies (Marzin and Braconnot, 2009; Goswami et al., 2006).

The large-scale spatial pattern of the summer monsoon is also different in EH compared to CTRL. Maps e) and f) on Figure 4 show the SLP anomalies and the location of the SLPa-5 barycentre. The EH depression extends further to the north-west and onto Africa and onto the Arabic peninsula than in CTRL and, the EH minimum SLP anomaly is moved to the north-west compared to the CTRL one (Fig.4e,f). The SLPa-5 barycentre, which is representative of the balance between the different
monsoons and of the Somali Jet position and direction, moves to the north-west in EH relative to CTRL (Fig4f). This suggests a modification of the monsoon structure with potential impacts on productivity through the atmospheric forcing onto the ocean.

Observation-based (Bauer et al., 1991; Prasanna Kumar et al., 2000) and model-based (Murtugudde et al., 2007) studies have shown that a stronger monsoon, with increased wind strength over the Arabian Sea, leads to higher productivity through intensified supply of nutrients in the photic zone. EH's Indian summer monsoon is enhanced compared to CTRL. One would
therefore expect the EH Arabian Sea productivity to be higher in EH than in CTRL. However, our model shows that the EH productivity is reduced in the Arabian Sea (Fig. 5a).



In this region, productivity is mainly nutrient-limited (Koné et al., 2009; Prasanna Kumar et al., 2000) and the levels of surface $NO_3^-$ concentration are lower for EH than for CTRL (Fig. 5b). Most of the nutrients are provided to the surface layer via Ekman dynamics: coastal upwelling, Ekman pumping and mixing of the upper layers close to the coast and offshore and, mixing and advection processes further offshore, in central Arabian Sea (Resplandy et al., 2011; Murtugudde et al., 2007; Prasanna Kumar et al., 2000; Lee et al., 2000; Bauer et al., 1991; McCreary et al., 2009). Figure 5c shows that wind stress intensity increases close to the Omani coast and decreases offshore. This leads to less mixing in the central Arabian Sea because of a shallower mixed layer (ML) (Fig.5d). The ML is deeper in EH than in CTRL close to the Oman coast (Fig.5d), which suggests a reduction in the upwelling intensity in EH. Wind stress intensity changes along with ML changes and the wind stress distribution of the CTRL simulation (Fig.3c) imply that the maximum wind intensity has moved closer to the Oman coast in the EH. This displacement of the jet reduces the wind stress intensity in the central sea. Moreover, the stronger EH wind stress very close to the coast (Fig. 5c) has a reduced effect on the ocean as it affects a smaller part of the coastal ocean. The reduction in coastal upwelling and mixing then contribute to the decrease in nutrients concentration in the upper layer (Fig.5b). The positive wind stress curl on the left-hand side of the jet has also a reduced influence onto the ocean (Fig.5e). The negative wind stress curl on the other side of the wind jet, affects a region closer to the coast leading to reduced Ekman pumping (Fig.5f). In summary, the EH atmospheric forcing through wind stress and wind stress curl is not able to generate as much coastal upwelling and Ekman pumping as in the CTRL simulation, even though the monsoon is stronger.

On Figure 6, we represented the 100 summers of the EH simulation minus the seasonal summer mean of the CTRL simulation for the variables TPP, wind stress and wind stress curl in the coastal and central Arabian Sea (regions on Figure 2a). In the coastal area, even though wind stress is always higher in EH (y-axis), productivity is always lower in EH than in CTRL (Fig.6a). This highlights the role of the wind stress curl that is always lower in EH than in CTRL (x-axis) : a smaller wind stress curl is responsible for lower productivity levels in the coastal western Arabian Sea. The wind stress intensity also affects the intensity of the changes in TPP: the stronger the wind stress the smaller the change in TPP. Wind stress can oppose the negative effect of reduced wind stress curl, but not overcome it as it is restricted very close to the coast (Fig. 5g).

In the central-east Arabian Sea, productivity is also lower in EH than in CTRL. The wind stress and the wind stress curl intensities are reduced almost every year compared to the CTRL seasonal summer mean (Fig.6b). This causes an increased downwelling and less wind mixing of the upper layers, which leads to reduced entrainment of nutrients and less productivity than in CTRL.

In summary, in the EH simulation the summer monsoon intensity is stronger than in the CTRL simulation but the productivity in the Arabian Sea is lower. This is caused by a shift in the Somali Jet position, which reduces coastal upwelling and Ekman pumping. This shift in the maximum wind intensity position closer to the coast can be inferred from the north-western movement of the SLPa-5 barycentre that translates into a modification of the monsoon pattern (Fig. 4).





## 3.2 Generalisation

In the previous section, we saw that a stronger monsoon in the EH does not imply more productivity in the Arabian Sea and that it is important to consider the spatial movements of the monsoonal winds. We now examine the links between productivity, monsoon intensity and boundary conditions in the remaining set of 6 glacial and inter-glacial simulations.

Figure 7 shows the changes in productivity, in the Arabian Sea, in all the remaining climates compared to the CTRL climate. Similar to the EH results, the MH, LGM, MIS4M and MIS4D coastal productivities are reduced (Fig. 7a-b,d-f). MH and MIS4F coastal productivity are reduced in average but present a dipole-like pattern, with higher productivity in the north and reduced productivity in the south compared to CTRL (Fig. 7a,d). Coastal productivity is enhanced in the MIS3 simulation (Fig. 7c). The central Arabian Sea productivity is higher than CTRL in all glacial simulations except MIS4F while it is reduced in MH

(and EH as seen previously).

    The tropospheric temperature gradient ($\Delta$TT) for each simulation, on Figure 8a, informs that the Indian summer monsoon intensity is stronger than CTRL in MH, EH, MIS3 and MIS4F (i.e. higher $\Delta$TT values) and less intense compared to CTRL in LGM, MIS4M and MIS4D. The changes in productivity for the western coastal Arabian Sea are also summarised on figure 8b. By only looking at these two variables, $\Delta$TT and TPP, we cannot conclude on a direct link between monsoon intensity

and productivity because stronger monsoons compared to CTRL, as characterised by $\Delta$TT, do not necessarily imply higher productivity (Fig. 8a,b), in particular for MH and EH.

### 3.2.1 Productivity and local dynamics

Productivity is nutrient-limited in the region and coastal productivity changes are similar to the changes in the nitrate content of the upper 30 m of the ocean (Fig. 8b,c). When the upper layer receives more nutrients from the subsurface, there is either

a stronger upwelling or a higher macro-nutrients ($NO_3^-$ and $PO_4^{3-}$) concentration under the mixed layer associated with enhanced entrainment and, productivity is higher than in CTRL. The stronger monsoon intensity, characterised by a higher $\Delta$TT value, is associated with higher values of coastal wind stress (Fig. 8a,d). But changes in wind stress curl are independent of the monsoon intensity since it is lower than CTRL in MH, EH and MIS4F and higher in all the other glacial simulations (Fig. 8e). Wind stress curl intensity is also higher in all the glacial climates compared to the Holocene. In MH, EH and MIS4F,

even though wind stress intensity is stronger and should generate more coastal upwelling, the highly reduced wind stress curl overcomes this positive effect on productivity and induces lower levels of macro-nutrients, which in turn limit productivity (Fig. 8b-e). Conversely, in LGM, MIS4M and MIS4D, even though the wind stress curl is higher than in CTRL, the lower wind stress intensity seems to prevail and productivity is reduced because of lower concentrations of nutrients (Fig. 8b-e). In MIS3, both the wind stress and the wind stress curl are more intense, more nutrients are brought to the surface and productivity

increases (Fig. 8b-e).

    Figure 9a summarises the links between productivity, wind stress and wind stress curl intensities in the coastal Arabian Sea. It shows that changes in wind stress and wind stress curl are drivers of productivity changes. Both variables modulate the change in productivity, with higher values associated with higher productivity.





In the central Arabian Sea, the enhanced productivity appears to result from more intense wind stress, which deepens the mixed layer and brings more nutrients to the upper layer through wind mixing and advection, consistent with Murtugudde et al. (2007), Wiggert et al. (2005) and Bauer et al. (1991). These results are summarised in Figure 9b, where the summer productivity changes are plotted as function of wind stress and wind stress curl summer changes in the central Arabian Sea.

When the wind stress intensity change is higher than in CTRL and wind stress curl not too low, productivity is enhanced (LGM, MIS4m and MIS4d)(Fig. 9b). If the wind stress is high enough it can compensate for a more reduced wind stress curl (MIS3) otherwise, productivity is lower (MIS4f, EH and MH)(Fig. 9b). Surprisingly, the simulations with a stronger summer monsoon do not systematically have stronger wind stress intensity over the central Arabian Sea but they have lower wind stress curl. This is the result of a change in the monsoon pattern, like a shift in the position or the orientation of the maximum wind intensity.

### 3.2.2  Relation to the large-scale forcing and boundary conditions

In order to understand how changes in the monsoon pattern are linked to the imposed boundary conditions and influence productivity and local monsoonal changes, we use the SLPa-5 barycentre. The position of the barycentre of each simulation is plotted on a map on figure10a. We also added on Figure 10, the mean boreal summer values (color-scales) of productivity, wind stress and wind stress curl as a function of the longitude (x-axis) and the latitude (y-axis) of the SLPa-5 barycentre.

Productivity shows an increasing trend with the longitude of the SLPa-5 barycentre between 70°E and 94°E (Fig.10b). It reaches a maximum of 30 $molC.m^{-2}.yr^{-1}$ around 94°E. For higher values of longitude, which correspond to simulations where the monsoon intensity is reduced, productivity decreases (Fig.10b). The higher values of productivity occur in the simulations for which the SLPa-5 barycentre's longitude and latitude have medium values (MH, CTRL, MIS4F and MIS3)(Fig.10b). The trends in productivity can be explained by the variations of wind stress (Fig.10c) and wind stress curl (Fig.10d) with the

SLPa-5 barycentre's position.

Wind stress exhibits a global decrease with the longitude and latitude of the SLPa-5 barycentre (Fig. 10c). If we except the CTRL simulation, wind stress is quite constant for the 5 first simulations (i.e. lower value of longitude) and then it decreases strongly for MIS4M, MIS4D and LGM. The wind stress shows a clear separation between simulations in which the monsoon is enhanced and wind stress is higher (EH, MH, MIS4F and MIS3) and those in which monsoon is less intense and wind stress

is reduced (i.e. LGM, MIS4D, MIS4M and CTRL)(Fig. 10c). Wind stress curl shows an increasing trend with longitude for the 5 simulations having lower values of longitude and then the wind stress curl becomes quite constant for MIS4M, MIS4D and LGM (Fig. 10d). The wind stress curl has also a tendency to decrease with the SLPa-5 barycentre's latitude (Fig. 10d).

The increase of productivity with longitude is mostly due to an increase in wind stress curl intensity and then the reduction of productivity with higher longitude is caused by a strong reduction in wind stress while the wind stress curl remains constant

(Fig.10b-d). These plots also show that all the glacial simulations have higher values of SLPa-5 barycentre longitude compared to CTRL. This suggests a major role of the ice sheet cover over the longitudinal position of the SLPa-5 barycentre. The simulations having a stronger monsoon intensity than CTRL have the highest value of SLPa-5 barycentre latitude, which suggests an influence of the orbital parameters on the latitudinal position of the SLPa-5 barycentre.



On figure 11, we plotted the values of the climatic precession and obliquity relative to the SLPa-5 barycentre position, in order to analyse the relationship between the orbital parameters and the SLPa-5 barycentre's position. Climatic precession influences the SLPa-5 barycentre position in longitude: when the climatic precession is high, the barycentre tends to move to the south-east (Fig.11a). Obliquity modulates the latitudinal changes of the SLPa-5 barycentre: high obliquities are associated with a SLPa-5 barycentre farther north (Fig.11b). MIS3 has a precession value similar to CTRL and a much higher obliquity than CTRL, so the changes in MIS3 winds and productivity related to insolation are mostly obliquity-driven (Fig. 11). Inversely, MIS4F has a similar obliquity as CTRL and a smaller climatic precession which implies that the changes in monsoon intensity in MIS4F are related to precession (Fig. 11). The Holocene simulations are influenced by both obliquity and precession while, the LGM, MIS4M and MIS4D seem to reflect a stronger link with the obliquity signal than with the climatic precession (Fig. 11).

## 4 Discussion

### 4.1 The summer monsoon paradigm

In the simulations, the general paradigm stating that a stronger summer monsoon intensity induces a stronger upwelling and therefore increases marine productivity, is not always verified. Our results show that the only characterisation of the summer monsoon intensity is probably insufficient to assess past productivity changes and reciprocally (Fig.8).

Our results for the summer productivity are consistent with the reconstructed productivity of Rostek et al. (1997). In their study, they analyse two cores in the Arabian Sea: one in the south-east (5°04' N - 73°52' E) and one in the upwelling region close to the Oman coast (13°42' N - 53°15' E). They show that paleo-productivity in the south-eastern core was higher in glacial stages than in interglacial stages, which they interpret as the fingerprint of a stronger winter monsoon. In the other core, the productivity signal is more complex and they could find some glacial stages (e.g. stage 2) with high productivity, some interglacial stages with low productivity (e.g. stage 1) and high productivity during stage 3. Similarly, in the simulations, in the central Arabian Sea, glacial productivity is higher than interglacial productivity (except for MIS4F) (Fig.9b). In the coastal Arabian Sea, the simulated MIS3 productivity is higher than CTRL while the other climates productivity is lower than CTRL. Hints on the sources of discrepancies between our results and Rostek et al. (1997) results and their interpretation of the productivity changes are given later in this section.

We explain the simulations' summer productivity changes by analysing the variations in the wind forcing (Fig.8). Given the productivity changes in the different simulations, the summer monsoon intensity only is not able to explain the changes in productivity and therefore, we also investigate the changes the monsoon pattern (Fig.10).

The large-scale definition of the monsoon intensity, via $\Delta TT$, is mainly driven by orbital changes. The simulations with a strong summer monsoon either have a high obliquity, which enhances the temperature contrast between low and high latitudes in summer (e.g. MIS3) or a small climatic precession that intensifies the summer insolation (MIS4F), or both (MH and EH) (Fig.11) (?). Simulations with a weak summer monsoon all have a small obliquity forcing (Fig.11b). The local wind stress that affects productivity is tightly coupled to the monsoon intensity (Fig.8a,b) and is associated with a SLPa-5 barycentre



movement to the North (Fig.10c). This simulated latitudinal movement of the SLPa-5 barycentre, according to the monsoon intensity, is consistent with the study of Anderson and Prell (1992). In this study, the authors show that with a stronger monsoon the Somali Jet is moved further North, which would indeed translate into a northward movement of the SLP barycentre (and inversely). In Fleitmann et al. (2007), the authors investigate the variations in the precipitation records of stalagmites located in

Oman and Yemen. They link the changes in precipitation to the position and structure of the ITCZ, which affects the tropical climate. They explain that during the early-Holocene, a northward movement of the mean latitudinal position of the summer ITCZ is responsible for the decrease in precipitation. Throughout the Holocene, they show that the ITCZ shifted southward concomitantly with a decrease in the monsoon precipitation, induced by the reduction of solar insolation (Fleitmann et al., 2007). The changes in the ITCZ position highlighted by Fleitmann et al. (2007) are consistent with our results, especially with

the changes in the position of the SLPa-5 barycentre in latitude (Fig.10).

      Marine productivity is not only influenced by the wind stress but also by the wind stress curl (Anderson et al., 1992) and the latter is also strongly influenced, in our simulations, by the glacial or interglacial state of the climate (Fig.10). The glacial-interglacial distribution of the wind stress curl is associated with a longitudinal movement of the SLPa-5 barycentre : the SLPa-5 barycentre is moved to the East in glacial climates and to the West in interglacial climates (Fig.10d). Pausata et al.

(2011) analyse the effects of different LGM boundary conditions on the atmospheric circulation and found that ice sheet to-pography is responsible for changes in many features of the SLP field, e.g. position of lows and highs and their variability. Consequently, changes in the ice-sheet cover between our simulations can indeed be responsible for the longitudinal movement of the SLPa-5 barycentre. Ivanova et al. (2003) also evoke an eastward shift in the low-level jet position in summer as a possible mechanism to explain some productivity changes in the eastern Arabian Sea. Furthermore, we showed that precession can also

act to move the SLPa-5 barycentre eastward (Fig.11a).

      We find a valid physical explanation for our simulations productivity changes through the effects of the simulations' boundary conditions (orbital parameters and ice sheets) on the monsoon intensity and pattern. However, even if the simulated summer productivity compares quite well to the data in Rostek et al. (1997) (except for the LGM), they do not match other reconstruc-

tions such as in Bassinot et al. (2011) (e.g. the Holocene productivity in the northern part of our coastal area). These differences can arise from several sources, the first one being linked to the area on which we computed our averages (Fig.2a). Indeed, in Rostek et al. (1997) the core in the upwelling region is taken in the southern part of our coastal area (Fig.2a) whereas the core close to the Oman coast in Bassinot et al. (2011) is located in the northern part of our coastal area (Fig.2a). Bassinot et al. (2011) reconstructed productivity is high in the early-Holocene and gradually decreases throughout the Holocene whereas, in

our simulations, the Holocene productivity is low compared to the pre-industrial productivity. A closer look at the productivity changes in the MH simulation on figure 7a can reconcile our simulations and this reconstructed productivity. In the simulated MH climate, the northern part of the coastal area exhibits a positive productivity change whereas, the southern part of the coastal area is characterised by a negative productivity change compared to CTRL. Therefore, our results are coherent with the results of Bassinot et al. (2011) for the mid-Holocene since their core is located in the northern part of our coastal area where

we simulate higher MH productivity than CTRL. However, we do not observe this dipole-like pattern in the EH productivity





(Fig.5a) and consequently, the EH simulation does not agree with this reconstruction. In the EH simulation, we do not have the remnant Laurentide ice sheet that is supposed to be present at this time-period (Licciardi et al., 1998). The addition of a remnant ice-sheet over Europe and North America in the EH has been shown, in Marzin et al. (2013), to induce a southward shift of the Inter-Tropical Convergence Zone (ITCZ) and a strengthening of the Indian monsoon. A southward shift of the jets

would modify the large scale pattern of the SLP and therefore, the wind stress and wind stress curl effects on the Arabian Sea. Based on our findings, the addition of this residual ice sheet would move the SLPa-5 barycentre to the south-east, which could increase the wind stress curl and therefore productivity (Fig.10).

Another source of mismatch between our simulations and data resides in the fact that we looked at productivity and not at the export production that will eventually reach the bottom of the ocean. We also focused the boreal summer season while it

is often advanced that the winter monsoon is responsible for higher productivity in glacial climates compared to interglacial climates, e.g. for the south-eastern core productivity in Rostek et al. (1997). In the next section, we especially discuss the effect of seasonality on productivity.

## 4.2 Seasonality

Here, we investigate new paleo-productivity reconstructions for the Arabian Sea and we compare them to our simulations.

The simulations globally agree with the reconstructions, with glacial productivities higher than Holocene productivities in north-western Arabian Sea, even during boreal summer.

We use data from the sediment core MD04-2873 located at 23°32N-63°50W in northern Arabian Sea on the Murray Ridge (Böning and Bard, 2009). This core is well dated by C14 dates from 50 kyr to present and has a marked Toba Ash layer (74 kyr BP, (Storey et al., 2012)) giving a significant robust stratigraphic marker. The coccolithophores are well preserved and abundant

at this location. Their assemblages are used to reconstruct paleo-productivity by using a transfer function that has been designed for the Indian Ocean including the Arabian Sea (Beaufort et al., 1997). Samples have been prepared by settling onto cover-slips (Beaufort et al., 2014) every 10 cm for stratigraphic intervals covering 2000 years above and below each time period simulated by the model. In average 6 samples were studied by time intervals. Coccolithophore analysis has been automatically generated by a software, SYRACO, that has been trained to recognise coccolithophores (Beaufort and Dollfus, 2004). Figure 12a shows

the resulting paleo-productivity for 7 of the 8 time periods we previously analysed. This reconstruction indicates that glacial productivity is higher than Holocene productivity in this core. At the core location, the effect of the boreal winter monsoon on productivity are known to be strong (Lévy et al., 2007). Consequently, the stronger winter monsoons during glacial time periods are often used to explain how glacial productivity can be higher than interglacial productivity (e.g. Rostek et al. (1997); Banakar et al. (2005)).

On figure 12, we also plotted the box-plots of simulated annual and summer (JJAS) productivity and export production at 100 m for the same time periods in the northern Arabian Sea (60°E-68°E, 20°N-68°N). Our simulated annual and boreal summer productivity show smaller differences between glacial and interglacial climates than the reconstructions (Fig.12a-c). The export production shows a clearer separation between the glacial and interglacial climates in both the summer and annual



plots (Fig.12d,e). The differences between productivity and export production (fig.12b-e) highlight that water column processes modify the recorded signal, which add difficulties when comparing model to data.

In all simulated climates, the mean boreal summer productivity is lower than the mean annual productivity (Fig.12). This indicates that, in this region, in the simulations, the mean boreal winter productivity is higher than the mean boreal summer productivity. Indeed, in the simulations, in the region of the core, boreal winter productivity accounts for more than 40% of the annual productivity and boreal summer productivity accounts for more than 20% (not shown). The hypothesis, stating that glacial productivity is higher than interglacial productivity because of a stronger winter monsoon, could explain the variations in this core. However, the observed present-day seasonal cycle of productivity in the northern Arabian Sea shows equal contributions of the winter and summer seasons to the annual productivity (Fig.2b), suggesting that the simulations may underestimate the boreal summer contribution to productivity compared to the boreal winter contribution.
Interestingly, the annual and boreal summer productivity plots look alike, with higher glacial than inter-glacial productivity (Fig. 12). Since the summer monsoon is able to affect the north-western Arabian Sea as seen on figure 2a, it contributes to the recorded signal in the sediment (e.g. Caley et al. (2011)). The boreal summer monsoon effect on the recorded signal is then non-negligible and, we see that, even during the boreal summer season, the simulations show higher glacial productivity than interglacial productivity (Fig. 12), especially in central Arabian Sea (Fig.9). Consequently, the boreal winter productivity is not the sole contributor to the higher glacial productivity signal compared to the interglacial productivity, even in the northern Arabian Sea.

## 5 Summary and perspectives

We use the coupled IPSL-CM5A-LR model to study the Arabian Sea paleo-productivity in 8 different climates of the past. We focus on the processes behind the boreal summer productivity changes in the coastal western and central-eastern Arabian Sea. We show that a stronger Indian summer monsoon, which is mostly driven by higher NH insolation, does not necessarily enhance the Arabian Sea productivity, and conversely.

We show that glacial climates can be more productive, in boreal summer, in the Arabian Sea (coastal and/or central), compared to the pre-industrial (Fig.9). Even more, the glacial climates are more productive than the early Holocene, which was supposed to be the most productive period in the region (Figs.9,12). We found that the paradigm between monsoon intensity and productivity is valid for MIS3, both in the coastal and central sea: a stronger monsoon leads to more productivity. The paradigm is also valid in the coastal Arabian Sea for the LGM, MIS4M and MIS4D simulations: a reduced monsoon intensity leads to a reduction in productivity. However it is not the case for MH, EH and MIS4F simulations for which a stronger summer monsoon is associated with reduced productivity, both in the coastal and central Arabian Sea. Moreover, the LGM, MIS4M and MIS4D simulations that have less intense summer monsoons have higher productivity levels in the central Arabian Sea.

Our analyses highlight the importance of considering the monsoon pattern, especially the position of the maximum wind intensity over the Arabian Sea. The mechanisms behind productivity changes are summarised on figure 13. We examine the monsoon pattern through the SLP barycentre position of the depression covering the summer monsoons regions (SLPa-5



barycentre). The SLPa-5 barycentre is moved to the East in glacial climates and far North in climates where monsoon is enhanced (Fig.10), which highlights the influence of the ice sheet cover (Pausata et al., 2011) and of the orbital parameters (Anderson and Prell, 1992). The monsoon pattern affects the wind stress and wind stress curl efficiency to bring more nutrients to the surface layers. A change in the pattern can reduce or increase the area on which the winds are effective. This study also

highlights the combined effects of wind stress and wind stress curl related processes on productivity. Neither wind intensity nor wind stress curl alone can explain productivity changes. We need to keep in mind that the model's coarse resolution does not allow for a very precise representation of the region dynamics. This may have altered the relative weight of the processes related to wind stress and wind stress curl and can explain why the orbital signal is weak in the productivity changes.

We demonstrated that both changes in wind stress and wind stress curl can affect productivity at the time-scales of thousands

of years (Fig.9). The same effects of wind stress and wind stress curl changes on productivity can be found at the inter-annual time-scale (Fig.14). Figure 14 illustrates the combined effects of wind stress and wind stress curl on productivity in the coastal Arabian Sea at the inter-annual time-scale. It shows that if the summer (JJAS) wind stress and wind stress curl intensities are higher than their summer average, productivity is higher than average in coastal Arabian Sea (upper-right quadrant). It also highlights that the higher the wind stress and the wind stress curl anomalies, the higher the productivity change, and conversely.

Figure 14 also shows that a high wind stress curl (resp., wind stress) can compensate a reduced wind stress (resp., wind stress curl) intensity and lead to higher than average productivity (lower-right and upper-left quadrants of figure 14, respectively). The relationship between changes in stress or curl and productivity is similar to the one we found for the glacial-interglacial climate changes (Figs.14 and 9). It could be interesting to further investigate these relationships by looking at high resolution models and re-analyses.

This study allows us to draw attention on certain points that may affect the reconstruction of past climate climate and productivity as well as the comparison between model and data. In addition, in regards of projected changes in the monsoon intensity and structure, these results can add some constraints on future productivity changes in the region. In chapter 14 of the 2013 IPCC report (Christensen et al., 2013), it has been shown, through the use of climate projections, that the future Indian summer monsoon should strengthen in regards of precipitation but become less intense in regards of the monsoon flow.

Moreover, Sandeep and Ajayamohan (2014) have shown that the projected low-level jet over the Arabian Sea will shift north because of global warming. A northward shift of the low-level jet is consistent with an increased monsoon intensity in our simulations. Then, if a stronger summer monsoon calls for increased productivity, a northward shift of the Somali Jet can either lead to reduced productivity, as in the Holocene and MIS4F simulations, or to an increased productivity as in the MIS3 simulation, depending on the degree of the shift and on the wind stress curl change.

*Acknowledgements.* Priscilla Le Mézo is funded by a grant from the Initiative D'EXcellence (IDEX) Paris-Saclay. This work was supported by the French ANR Project ELPASO (No.2010 BLANC 608 01), the computing time was provided by GENCI (Grand Equipement National de Calcul Intensif) and the simulations were performed on Curie at TGCC (CEA, France).





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



**Table 1.** Boundary conditions for the 8 simulations studied in this work. Precession is defined as the longitude of the perihelion, relative to the moving vernal equinox, minus $180°$. Ice sheets are represented on figure 1. "Pmip3" ice-sheet stands for the PMIP3 ice sheet reconstruction (Abe-Ouchi et al., 2015), ICE6g-16k stands for the ICE6G reconstruction at 16 kyr BP (Peltier et al., 2015)

| | Interglacial climates | | | Glacial climates | | | | |
|---|---|---|---|---|---|---|---|---|
| **Simulation name** | **CTRL** | **MH** | **EH** | **LGM** | **MIS3** | **MIS4F** | **MIS4M** | **MIS4D** |
| **Time (kyr BP)** | 0 | 6 | 9.5 | 21 | 46 | 60 | 66 | 72 |
| **Ice sheets** | present | present | present | pmip3 | Ice6g-16k | Ice6g-16k | Ice6g-16k | Ice6g-16k |
| **Sea level difference vs CTRL (m)** | 0 | 0 | 0 | -120 | -70 | -70 | -70 | -70 |
| **Eccentricity** | 0.016715 | 0.018682 | 0.0193553 | 0.018994 | 0.0138427 | 0.018469 | 0.021311 | 0.024345 |
| **Obliquity ($°$)** | 22.391 | 24.105 | 24.2306 | 22.949 | 24.3548 | 23.2329 | 22.493 | 22.391 |
| **Precession ($\omega - 180°$)** | 102.7 | 0.87 | 303.032 | 114.42 | 101.337 | 266.65 | 174.82 | 80.09 |
| $CO_2$ **(ppm)** | 284 | 280 | 284 | 185 | 205 | 200 | 195 | 230 |
| $N_2O$ **(ppm)** | 275 | 270 | 275 | 200 | 260 | 230 | 217 | 230 |
| $CH_4$ **(ppb)** | 791 | 650 | 791 | 350 | 500 | 426 | 450 | 450 |



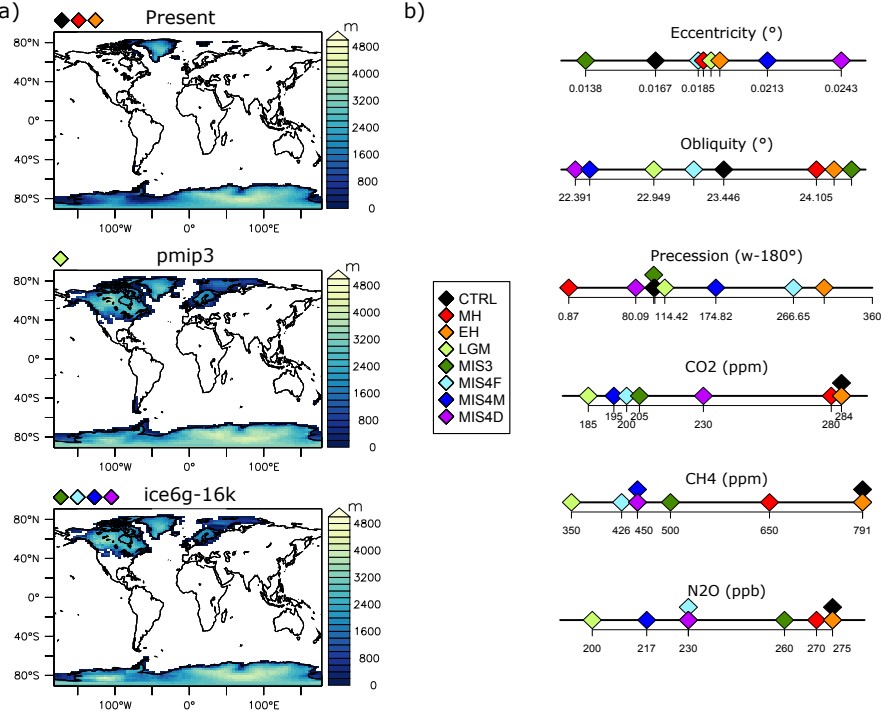

**Figure 1.** a) The three ice sheet covers used as boundary conditions. The 0k ice sheet is used in the CTRL, MH and EH simulations, the pmip3 ice sheet (Abe-Ouchi et al., 2015) is used in the LGM simulation and the ICE6g-16k ice sheet (Peltier et al., 2015) is used in the MIS3 and all the MIS4 (F, M and D) simulations and b) the different values of the other forcing parameters for all the simulations.

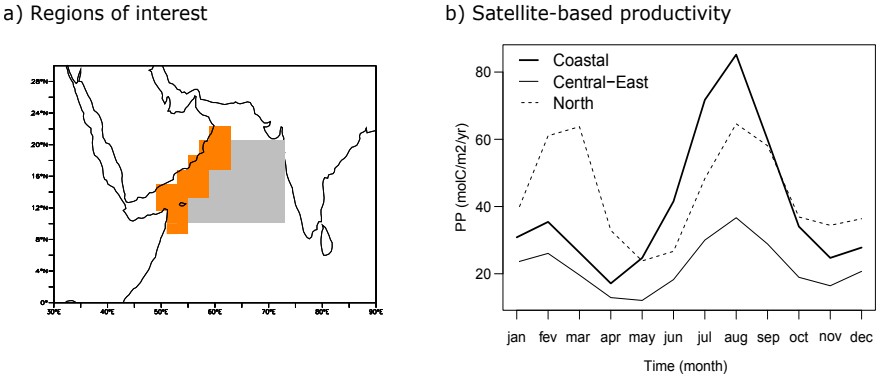

**Figure 2.** a) Coastal (orange shading) and central-east (grey shading) Arabian Sea areas and b) Seasonal cycles of productivity in the coastal (bold line), central-eastern (simple line) and northern (60°E-68°E, 20°N-68°N, dashed line) Arabian Sea.





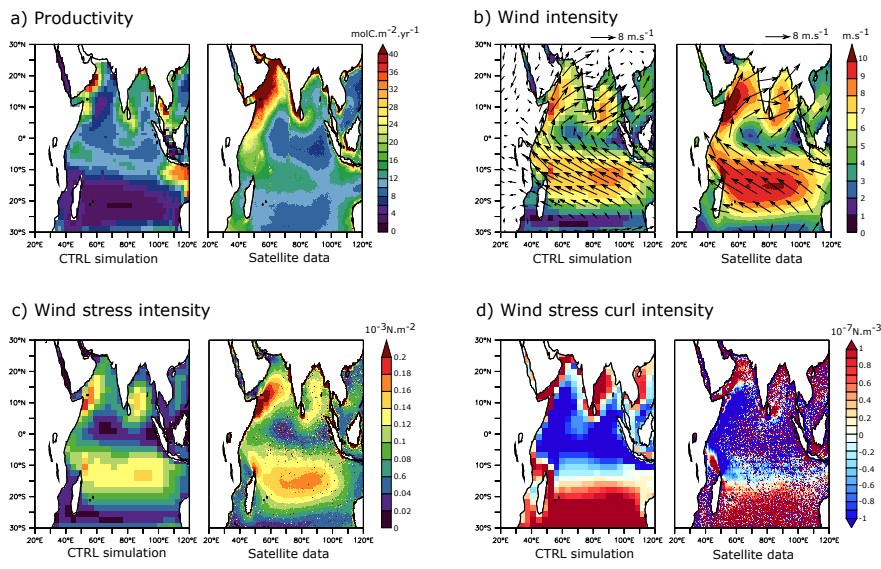

**Figure 3.** Modelled and observed seasonal (JJAS) patterns of a) productivity $(\mathrm{molC.m}^{-2}.\mathrm{yr}^{-1})$, b) surface wind intensity $(\mathrm{m.s}^{-1})$, c) surface wind stress intensity $(10^{-3}\mathrm{N.m}^{-2})$ and d) surface wind stress curl intensity $(10^{-7}\mathrm{N.m}^{-3})$. We used SeaWIFS data in 1998-2005 for productivity (Lévy et al., 2007) and the NOAA Multiple-Satellite 1995-2005 product (Zhang, 2006) for the climatology of surface wind, wind stress and wind stress curl intensity.



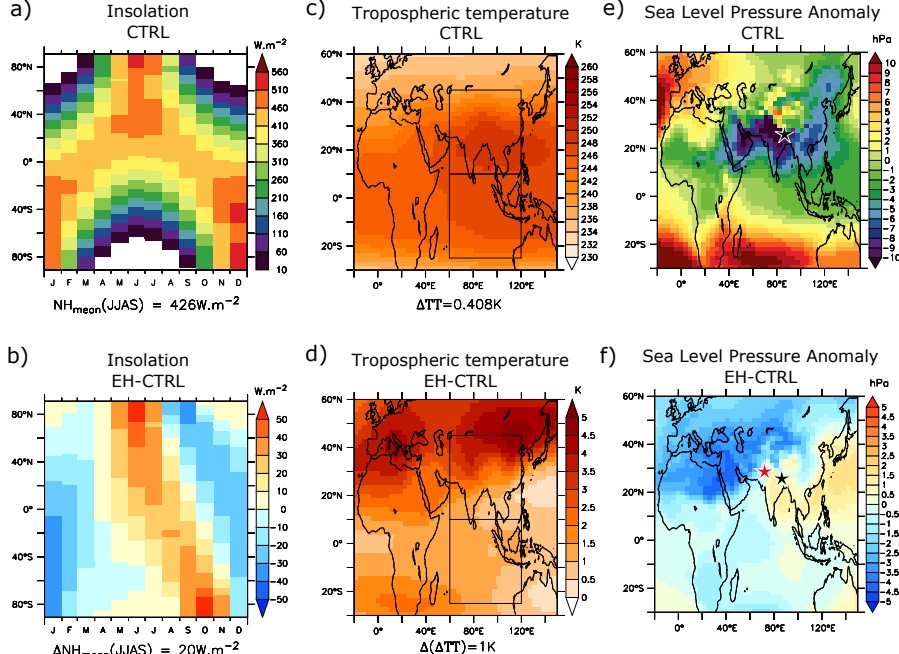

**Figure 4. a,b** : Seasonal cycle of the insolation at the top of the atmosphere (W.m$^{-2}$) in a) the CTRL simulation and b) the difference between EH and CTRL. The Northern Hemisphere (0-90°N) mean summer (JJAS) insolation, NH$_{mean}$(JJAS), in CTRL and the difference between EH and CTRL, $\Delta$NH$_{mean}$(JJAS), is given under panel a) and b), respectively.

**c,d** : Upper tropospheric temperature (TT) averaged between 200 hPa and 500 hPa in c) CTRL and d) the difference between EH and CTRL. $\Delta$TT value, under panel c), is the TT gradient between a northern region (60°E-120°E; 10°N-45°N) and a southern region (60°E-120°E; 25°S-10°N) (black boxes on the maps). $\Delta(\Delta TT)$, under panel d), is the difference of TT gradients between EH and CTRL.

**e,f** : Boreal summer (JJAS) sea level pressure (SLP) anomaly (from the annual mean) for e) CTRL and f) EH-CTRL. The SLPa-5 barycentre (i.e. barycentre of the SLP anomalies lower than -5 hPa over the region 20°W-150°E; 30°S-60°N) is represented by a black star for CTRL on e) and f) and by a red star for EH on f).




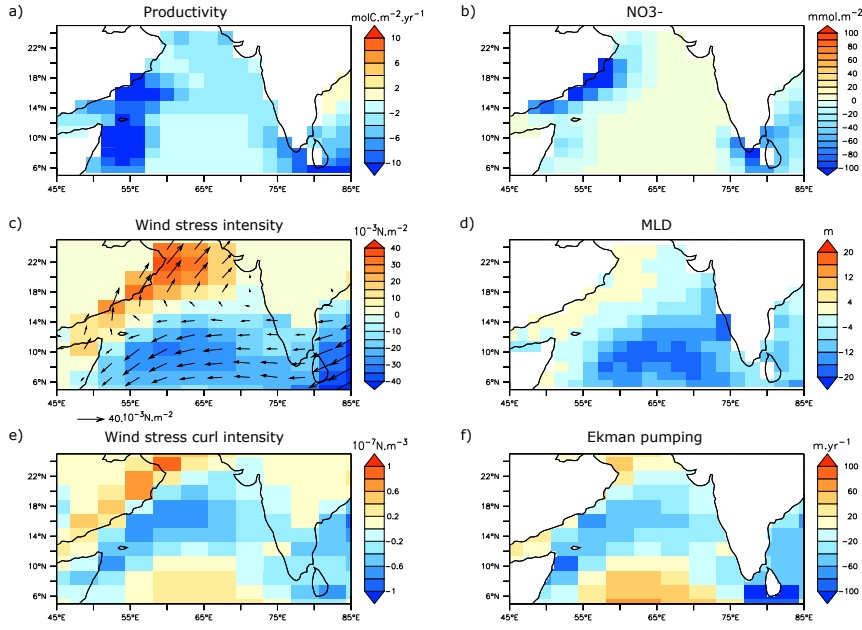

**Figure 5.** Boreal summer mean differences between EH and CTRL for a) total primary productivity (TPP, $molC.m^{-2}.yr^{-1}$) integrated over the whole water column, b) $NO_3^-$ concentration ($mmolC.m^{-2}$) in the first 30 m of the water column, c) wind stress intensity ($N.m^{-2}$) and direction (arrows), d) mixed layer depth (MLD, m), e) wind stress curl intensity ($10^{-7}N.m^{-3}$) and f) Ekman pumping ($m.yr^{-1}$).

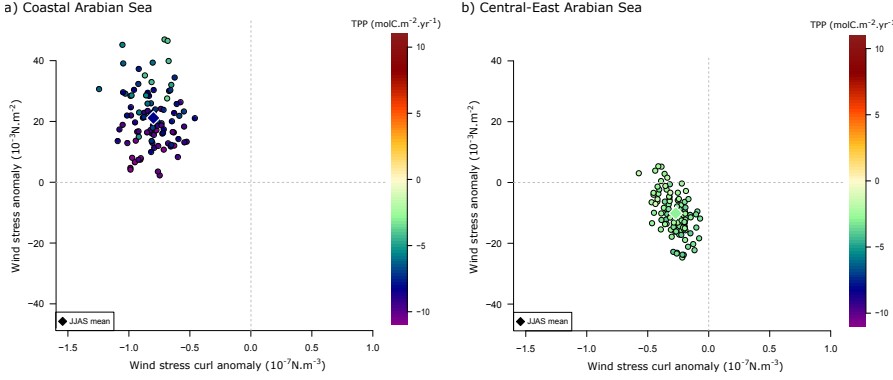

**Figure 6.** Seasonal (JJAS) anomalies of total primary productivity (TPP, $molC.m^{-2}.yr^{-1}$), integrated over the whole water column, as a function of wind stress anomalies ($10^{-3}N.m^{-2}$) and wind stress curl anomalies ($10^{-7}N.m^{-3}$) in a) the coastal western Arabian Sea and b) the central-east Arabian Sea (see Fig. 2a). Anomalies are computed as the difference between each yearly summer average in the EH simulation and the seasonal summer mean of the CTRL simulation. The color of the circles represents the value of the change in TPP between EH and CTRL. The color-scale, x-axis and y-axis ranges are the same as those in Figure 9 and Figure 14.



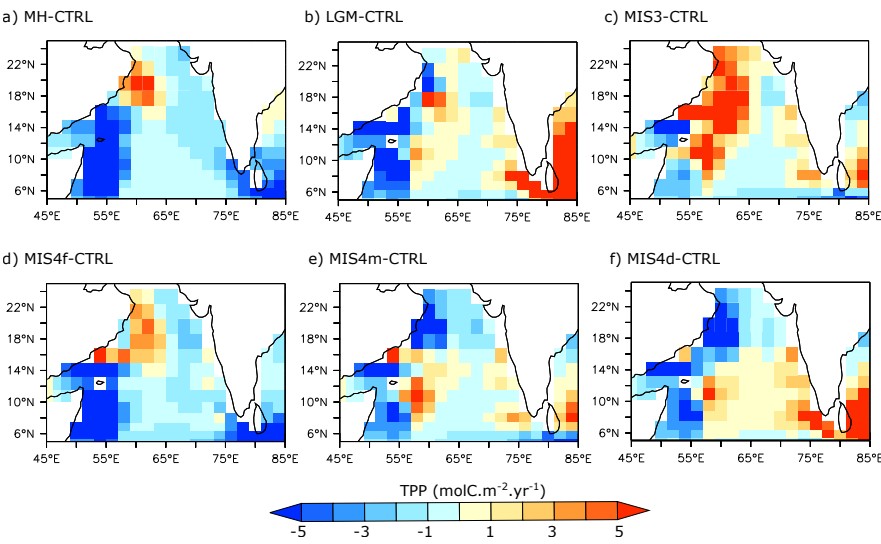

**Figure 7.** Seasonal (JJAS) productivity changes $(\mathrm{molC.m^{-2}.yr^{-1}})$ compared to the CTRL simulation for a) MH, b) LGM, c) MIS3, d) MIS4F, e) MIS4M and f) MIS4D simulation in the Indian Ocean.

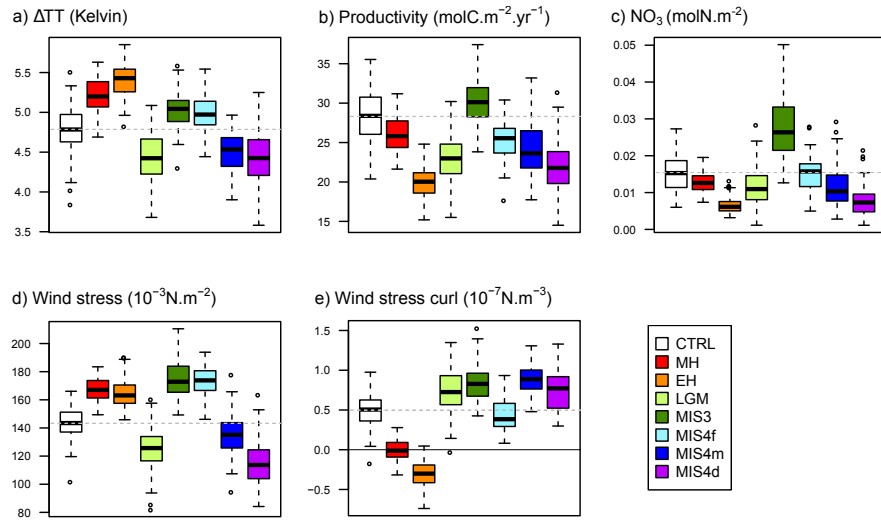

**Figure 8.** Box-plots of summer (JJAS) a) $\Delta TT$ (K) values, coastal Arabian Sea b) productivity $(\mathrm{molC.m^{-2}.yr^{-1}})$ integrated over the whole water column, c) nitrate concentration $(\mathrm{mmolC.m^{-2}})$ in the first 30 meters, d) wind stress intensity $(10^{-3}\mathrm{N.m^{-2}})$ and e) wind stress curl intensity $(10^{-7}\mathrm{N.m^{-3}})$, for all 8 simulations. Dash grey line indicates the CTRL value for each variable. The black simple line on e) panel indicates the zero value.





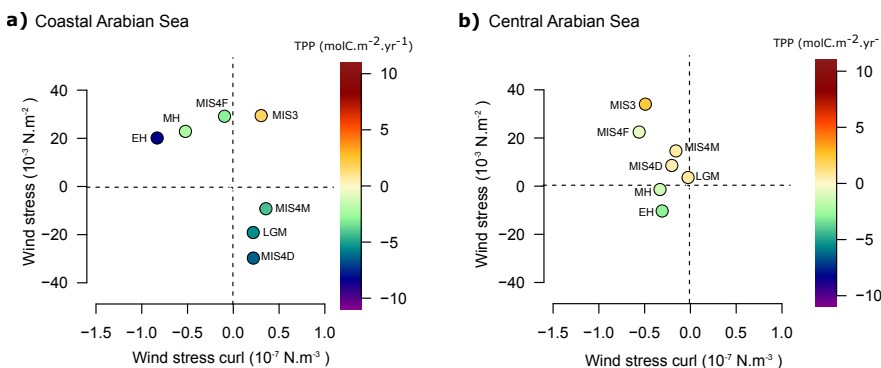

**Figure 9.** Seasonal (JJAS) productivity changes ($\mathrm{molC.m^{-2}.yr^{-1}}$) related to wind stress intensity ($10^{-3}\mathrm{N.m^{-2}}$, y-axis) and wind stress curl intensity ($10^{-7}\mathrm{N.m^{-3}}$, x-axis) changes compared to the CTRL simulation, for a) coastal and b) central Arabian Sea. The color scale, x-axis and y-axis ranges are the same as those in Figure 14 and Figure 6.





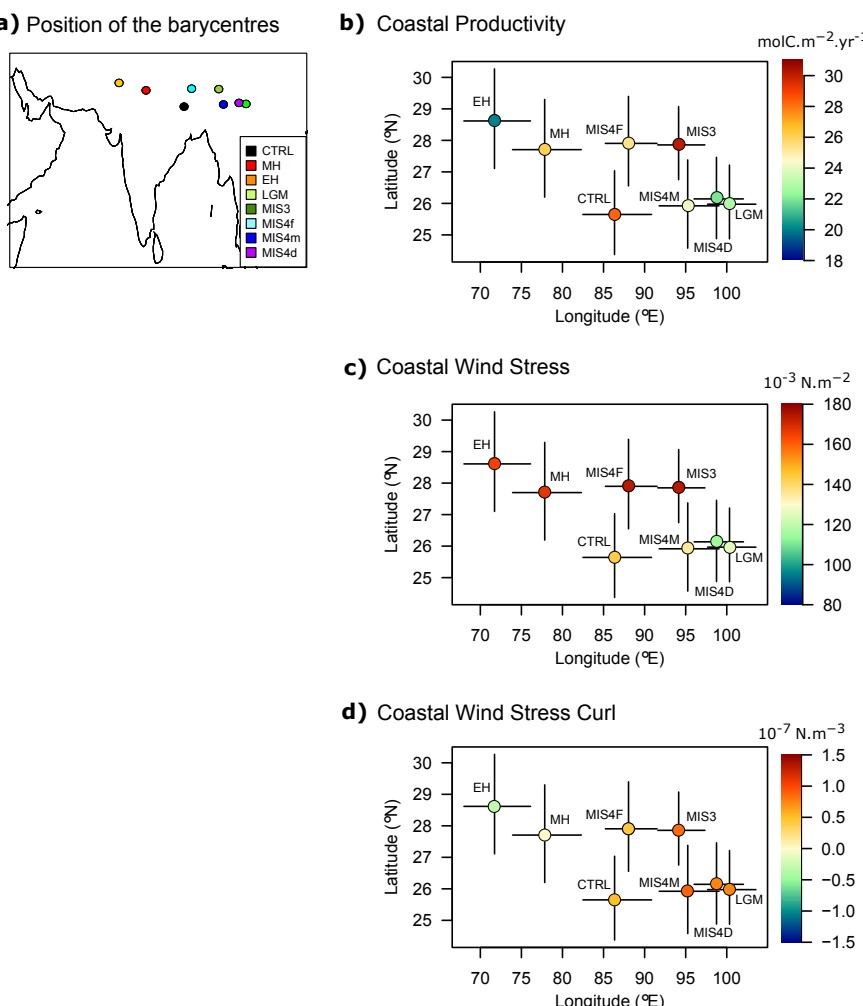

**Figure 10.** Seasonal (JJAS) coastal productivity ($molC.m^{-2}.yr^{-1}$), wind stress intensity ($10^{-3}N.m^{-2}$, y-axis) and wind stress curl intensity($10^{-7}N.m^{-3}$, x-axis) as a function of the SLP barycentre a-c) longitude and d-f) latitude. Errors bars give standard deviation of the 100 summers.




**a)** Climatic precession

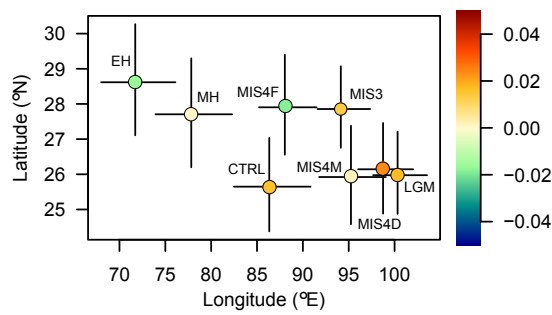

**b)** Obliquity

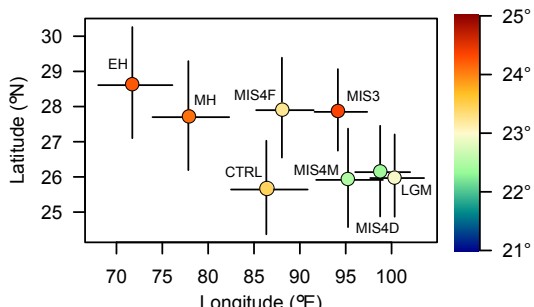

**Figure 11.** Mean a) climatic precession (e*sin($\omega - 180°$) where e is the eccentricity and $\omega - 180°$ is the precession) and b) obliquity values as a function of the longitude and latitude of the SLPa-5 barycentre for the 8 climate simulations. Errors bars give standard deviation of the SLPa-5 position over the 100 summers of each simulation.





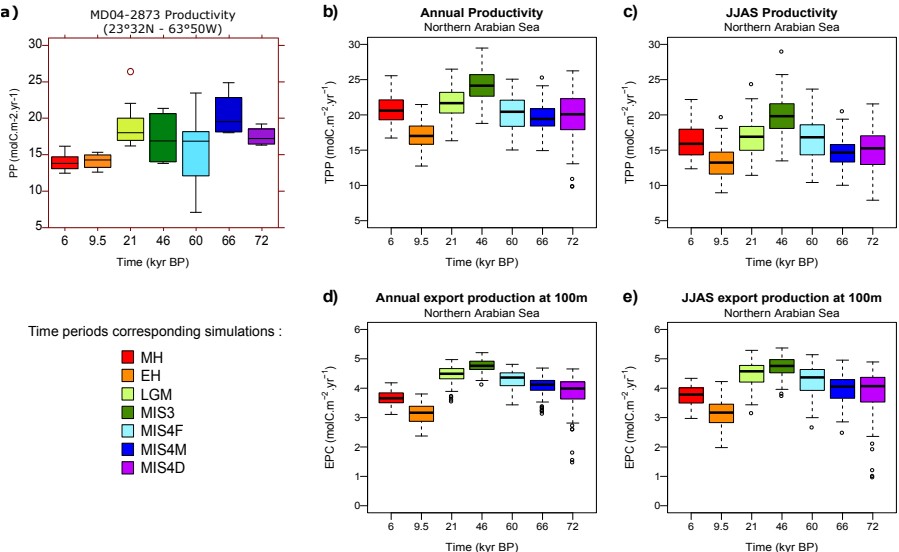

**Figure 12.** Box-plots of a) yearly reconstructed productivity from core MD04-2873 in the north-western Arabian Sea and, b) annual total primary productivity (TPP), c) summer total primary productivity, d) annual export production (EPC) at 100 m and e) summer export production at 100 m, in the northern Arabian Sea (60°E-68°E, 20°N-68°N) for the Holocene and glacial time periods.

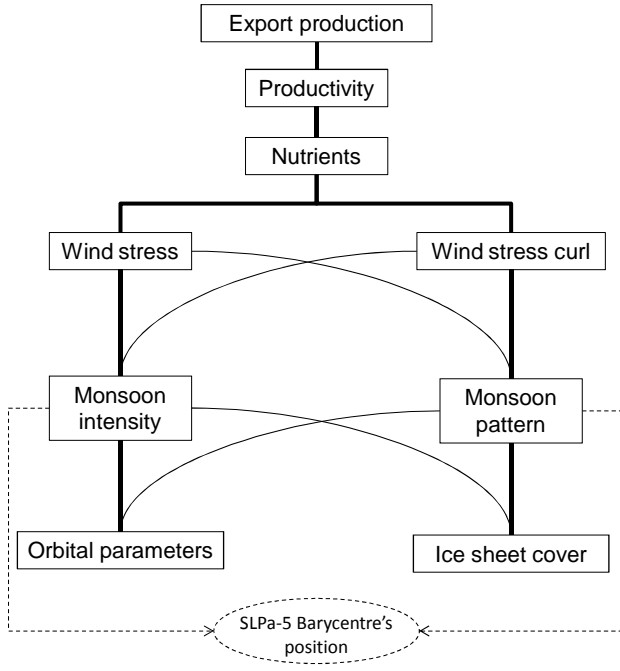

**Figure 13.** Identified seasonal (JJAS) processes behind productivity changes in glacial and inter-glacial climates. Bold lines highlight the major pathways.



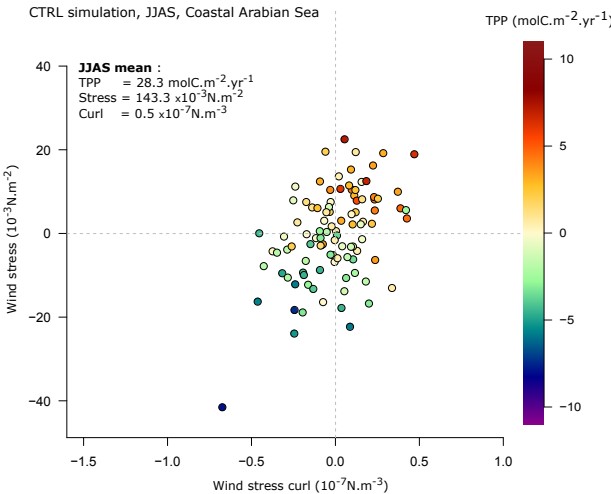

**Figure 14.** Anomalies of total primary productivity (TPP, $\mathrm{molC.m^{-2}.yr^{-1}}$, represented by the color-scale) integrated over the whole water column as a function of wind stress anomalies ($10^{-3}\mathrm{N.m^{-2}}$, y-axis) and wind stress curl anomalies ($10^{-7}\mathrm{N.m^{-3}}$, x-axis) in the coastal western Arabian Sea (see Fig. 2a) for the CTRL simulation. Anomalies are computed as the difference between each summer average (JJAS) and the seasonal summer mean of the corresponding variable in the CTRL simulation. The circles color represents the value of the change in TPP compared to CTRL. The color-scale, x-axis and y-axis ranges are the same as those in Figure 6 and Figure 9.