# Peer review of "From Monsoon to marine productivity in the Arabian Sea: insights from glacial and interglacial climates"

_Climate of the Past, 2016_

## Referee Comment (RC1) · Anonymous Referee #1 · 8 Dec 2016

Recommendation This paper investigates how changes in orbital parameters and ice sheets during Âăthe last glacial-interglacial cycle impact Arabian Sea productivity through changes in the monsoon intensity and spatial pattern. This is an important topic, since productivity estimates from Arabian Sea cores are usually interpreted as proxies of the monsoon intensity. Using numerical simulations with a earth system model, this study shows that the relationship between monsoon intensity and productivity is non trivial, because spatial shifts in the monsoon jet axis influence both wind stress and its curl, which both control the influence of the coastal upwelling. The paper topic is interesting, and the analyses are scientifically sound. I however suggest a couple of additional diagnostics, re-arangement of some results, additional critical dis-

cussion of the model, which should improve the clarity of the paper and make it more convincing (see general comments below).

General comments 1. I feel that some diagnostics on the intensity of the upwelling (eg SST anomaly in the coastal box relative to the Indian Ocean average and/or vertical velocities and/or depth of the thermocline – and nutricline-) would be helpful to relate the changes in nutrients with changes in the upwelling intensity. A bit more validation of the upwelling characteristics in the present-day simulation also would not harm. 2. There is no discussion of how the biases of the model for the current-day climate (e.g. its underestimated productivity) may influence the overall results of the study. 3. The abstract could be improved (suggestions below). It may also be beneficial to show and discuss Fig. 14 much earlier in the paper, in order to describe the relative effects of alongshore stress and near-shore Ekman pumping in the present-day climate ahead of the past climates discussion. 4. I am not sure that it is worth discussing the central Arabian Sea region (Figures 6 and 9). A reason for that is that this region is usually viewed as highly influenced by what happens in the coastal upwelling region. I hence feel that focussing on the coastal box is enough. If you want to keep this central Arabian Sea box, a diagnostic as that of figure 14, which relates the interannual variability of productivity to wind stress (here an indicator of vertical mixing) and wind stress curl would be helpful. 5. It may be useful to show maps of the JJAS climatological wind stress and Ekman pumping values for all the simulations, to more visually relate how the changes in low pressures over the continent relate to changes in the monsoon flow.

Detailed comments P1, L1: maybe "the current-climate Indian monsoon..." P1, L5-10: I think that the abstract could be clarified. Maybe mention explicitly that coastal upwelling is fuelled by a combination of alongshore stress intensity and upward Ekman pumping to the west of the jet axis. There is however strong downward Ekman pumping to the east of the jet axis, so that changes in coastal alongshore stress / curl depend both on the jet intensity and position. You can then relate changes in the intensity and position to the exact position of the low pressure over Tibet, with astronomical parameters having impact mostly on the intensity and changes in ice sheets rather influencing the jet position. P1, L18: Another useful reference here is McCreary et al. (2009) (see full reference below). P1, L24: maybe indicate "upward" Ekman pumping. It may be interesting to mention offshore downward pumping to the right of the jet axis. P2, L5: Also mention the specific role of eddies, quoting Resplandy et al. (2011). P2, L9-11: Recent studies indicate that, due to changes in atmospheric stability, an increase in rainfall is not necessarily associated with an increase of the associated circulation (e.g. Held and Soden J. Clim. 2006 in the context of anthropogenic climate change). P2, L24: Be more specific: change of ice sheet heights in which regions? Same comment for ice volume L27. P3, L14 and following: "the LMDZ5A atmospheric general circulation model" (and likewise for the other components) P4, L30-34: A recent study (Keerthi et al. 2016, see below for full ref) shows that, in winter (when the cloud cover over the Arabian Sea is low), there is a good agreement between various satellite datasets for the Northern Arabian sea, but large differences in terms of amplitude. You may hence want to add a cautionary note about uncertainties of the observational estimate. P5, L1: vertical mixing is more specifically due to convective overturning in presence of strong southward winds that bring dry, cold continental air. P5, L11: typo: ? -> '. How is productivity computed from SeaWifs? P5, L18-19: It is also likely that the absence of eddies in the model solution contributes to a weaker offshore export than in observations (but you note it a bit later on p6). The consequences of this underestimated productivity on your results has to be thoroughly discussed at the end of the paper. P5, L25-35: Time series of the mean seasonal cycle of alongshore wind-stress, near-shore wind stress curl and of an indicator of the upwelling (e.g. SST) would allow a more quantitative validation of the model than the existing figures. P6, L10: Is "pathway" appropriate in this context? P6, L23-25: You should refer to fig4 to justify this diagnostic better. I am surprised by the position of the black start on figure 4e: it is on the edge of the region with SLP anomalies < -5 hPa, which is surprising for a barycentre. Is there a gridpoint with a very large negative SLP anomaly? Or maybe I'm colorblind: you should probably highlight the -5 hPa contour on that figure. P8, L5-16: I generally agree with

the interpretation, but I feel that the explanation takes twists and turns. I would reorganize as follows: the shift of the Tibetan low leads to a poleward shift of the monsoon jet (5c). This leads to weaker alongshore winds in the Somalia and stronger alongshore winds in the oman upwelling (5c). But this also brings the Ekman downwelling to the right of the jet axis closer to both Oman and Somalia coasts (5e). Both factors (alongshore, and offshore curl) contribute to a Somalia upwelling reduction, while the wind stress curl change seems to overwhelm the increased alongshore winds in the Oman region, leading to overall upwelling reduction and less nutrients (5b). The reduced wind stress also reduces the mixed layer depth in the Southern Arabian Sea (5d) but I'm not sure it's so relevant to show this here. Rather, showing SST (which characterizes the upwelling intensity) and/or vertical velocities in the model would allow to better characterize physical changes. Showing the depth of the nutricline would also be really helpful. Finally, I wonder why nutrient reduction is larger in the Oman region than in the Somalia region. Is it an effect of changes in biological uptake? P8, L17-23: This figure is relevant for the coastal box, but is it so relevant for the offshore box, for which a lot of the productivity changes are most likely a consequences of changes in the coastal upwelling, exported offshore by the circulation. For the coastal box, it indicates that the curl change (favouring downwelling) wins over the change in coastal winds (favouring upwelling and enhanced vertical mixing). This would be confirmed by looking at the thermocline (and nutricline) depth and/or to vertical velocities, i.e. looking at physical indicators of the intensity of the upwelling. Another interest of this plot relies on the fact that productivity is decreased for all years: i.e. the change that you see is statistically significant. Finally the last interest of this plot is that it allows to evaluate the respective roles of Ekman pumping and alongshore wind stress for interannual variability: although not very clear with the current choice of color scale for panel a, there is a tendency for a weaker decrease in productivity for strong alongshore stress, as would be expected. We would also expect to see a stronger decrease in productivity for larger negative wind stress curl anomaly. If you confirm this (i.e. by performing a bilinear fit of the productivity on the stress & curl), illustrating the competition between those two

effects for interannual variability will strengthen your case for explaining changes between CTL and EH. Nota: I later came to realize that you actually show this as figure 14, but it may be interesting to show it at an earlier stage in the paper. P9, L9-10: This statement is more confusing than helpful (where exactly in the Central AS?) P9, L11: Figure 8 would be usefully complemented by scatterplots between some of the variables (and the associated correlations): alongshore wind stress is strongly controlled by DTT (panels a and b); nutrients exert a (weaker) control on productivity (b and c): I would combine these scatterplot with Figure 9a (again, I don't think that figure 9b is so relevant, because wind stress is not really an upwelling driver away from the coast and productivity changes in the central Arabian Sea may not be the result of local processes). P9, L24-25: Doing a scatterplot of alongshore wind stress vs. coastal wind stress curl would also be helpful to characterize the relations of the two parameters that control the upwelling. I would not say they are entirely independent: it seems that there is a tendency for a negative correlation. P10, L15-20: It may be better to start by explaining the links between the low pressure position and the wind stress and curl, and then discussing the consequences on the productivity. To understand better the links between the low pressure position and wind stress intensity / curl, it would be good to add a figure with maps of the JJAS wind stress (vectors) and its curl (colors), with the position of the barycentre indicated. P10, L23: simply point out that the latitude of the barycentre exerts a strong control on the coastal wind stress amplitude. P10, L28: In general, in this section mention "increase of productivity with the central longitude of the Tibetan low". P11, L1-10: I'm not that familiar with those orbital parameters. Wouldn't it be simpler to directly relate the position of the low pressure to the annual (or JJAS) solar heat flux properties over the region, and then describe more qualitatively how orbital parameters control this value? Or at least remind how these parameters influence insolation and hence the low (e.g. copy the text at lines 30-31 here). P11, L14: Âăremove only P13, L8-9: can you explain this choice? P13, L17: you could locate this core, e.g. on figure 7. P14, L9: provide a reference in support of this statement. P14, L11-17: in the model. P15, L9-19: This is nice. I would move this much earlier in the

paper, because it allows to explain better the relative roles of wind strees and its curl at interannual timescales before moving to applying these explanations for climates of the past. P15: a through discussion on the possible consequences of the present-day climate biases in the model on the results of this study is needed.

Figures Fig. 2: also draw the Northern box. Fig 3: left panels of b, c, d have some spurious horizontal lines on them. Some smoothing or median filter would not harm on the sat. wind stress intensity & curl. On d, also avoid saturating the colorscale. Panel b: use the same spacing between vectors for the model and data. Fig 5: it would be useful to materialize the boxes that you use for integrated diagnostics on this figure. Fig 7: draw the coastal box. Figure 8: describe the whiskers in the caption. I imagine it is a confidence interval: at what significance level? How is it computed? What are the small circles on this figue (and on figure 12). Figure 10: draw the boundaries of panels b, c, d on panel a. You can re-organize the panels into 2 x 2 to save space. Figure 12: it would be nice to locate the core / box that are used for figure 12 on another plot, e.g. figure 7. Clarify in the caption which panels are model results. Figure 13: I don't find this picture very clear. I would do it upside down: start it by changes in orbital parameters / ice sheet driving changes in the Tibetan plateau low, which impact the intensity / pattern, stress / curl, the upwelling of nutrients, etc. . . Materiaize the direction of arrows connecting boxes.

References McCreary, J. P., R. Murtugudde, J. Vialard, P. N. Vinayachandran, J. D. Wiggert, R. R. Hood, D. Shankar, and S. R. Shetye, 2009: Biophysical processes in the Indian Ocean, In: Indian Ocean Biogeochemical Processes and Ecological Variability, J.D.Wiggert, R.R. Hood, S.W.A. Naqvi, S.L. Smith, and K.H. Brink (ed.), American Geophysical Union, Washington, D. C, pp 9-32. Keerthi et al. 2016: Physical control of the interannual variations of the winter chlorophyll bloom in the northern Arabian Sea. Biogeosciences discussions, available at http://www.biogeosciences-discuss.net/bg-2016-153/

---

## Referee Comment (RC2) · Anonymous Referee #2 · 24 Dec 2016

The manuscript 'From Monsoon to marine productivity in the Arabian Sea: insights from glacial and interglacial climates' by P. Le Mézo et al. is an interesting model study on the driving forces of ocean productivity in the Arabian Sea. A large number of paleo climate simulations of different warm and cold climates, forced by changes in orbital parameters, greenhouse gases and northern hemisphere ice sheets is used to systematically investigate changes in productivity patterns. Against the common paradigm that there is a straightforward relationship between pure Monsoon strength (wind mixing) and marine productivity, which was often used to infer past Monsoon changes from productivity proxies, the study highlights important aspects of changes in the Monsoon pattern and potentially opposing regional trends, for example in coastal

and open ocean areas. According to the model results, wind stress curl, the changes of which are mostly driven by changes in the Monsoon pattern, turns out to be an equally important factor influencing productivity. The study is well designed, straightforwardly carried out and the demonstrated results are robust and clearly illustrated in many figures. The text could benefit from a number of clarifications, however, this does not compromise the overall excellent quality of the study. I suggest publishing after some minor revisions.

Major comments/questions: Since many readers are maybe not too familiar with the theory of wind stress curl, a short introduction into the most important mechanisms of upward nutrient transport and their interplay would probably be helpful.

Apart from MIS3, the control simulation stands out as the most productive period for the Arabian Sea. Can the results be extrapolated to other ocean areas, for example the prominent eastern boundary upwelling regions (EBUs)? What would be the consequences for the underlying oxygen minimum zone?

What are the model limitations? Does spatial resolution play a role? Since a rather coarse resolution was used here, what are the uncertainties and/or probable effects of parameterizations of sub gridscale processes or why would all these not matter? A first tentative explanation is given on page 15, l. 6-8, but this could be further elaborated.

When describing changes in wind stress curl often the words higher or lower are used, which may cause misunderstandings. Although the relative effect of the curl tendency on upward nutrient transport is the same, it needs to be kept in mind that the change maybe associated with a change of sign of the curl or not. I'd therefore suggest that the use of more or less negative/positive curl is a more correct wording. Please check throughout the text.

Minor points: p. 1, l. 6-8: This sentence is confusing. It seems that the combination of increased wind stress and more positive curl could result in a reduction of productivity. However, this is not the case (Fig. 9). I think it is more correct to state that both changes

in wind stress and curl need to be taken into account and that under opposing changes one or the other may be the ultimately dominating effect. p. 1, l. 12: typo, remove 's' in 'affects' p. 1, l. 21: replace 'in India' by 'over India' p. 3, l. 15: add 'semi-labile' before 'dissolved organic carbon' p. 3, l. 29: add 'greenhouse' before 'gas' p. 3, l. 30: add 'Northern Hemisphere' before ice sheets (at least from Fig. 1 this seems to be the case) p. 4, l. 8: This sentence is confusing. Do you count MIS3 as glacial or interglacial? If glacial then you should mention that five glacial simulations were carried out. Otherwise, it is strange to see it listed among the glacial climates. Please check. p. 4, l. 9: remove 'for' p. 4, l. 26: a few more information on length of simulation and starting conditions would be nice p. 4, l. 31: I guess that 68°N should be 28°N p. 4, l. 33: replace 'sediments bulk' by 'bulk sediment' p. 4, l. 34: replace 'changes' by 'peaks' and 'ones' by 'maxima' p. 4, l. 34: Does the sentence starting here refer to the winter period? If yes, then p. 5, l. 1: add 'dashed line in' before 'Fig. 2b) p. 5, l. 11: typo: NASA's p. 5, l. 16: 'Arabian' instead of 'Arabic' p. 5, l. 29: what do you mean by reconstructions? Observations? Please explain. p. 6, l. 1: move 'coarse' before 'model' p. 6, l. 6 'model's' p. 6, l. 20: in several places of the manuscript I'm wondering about the use of the word 'global'. Do you really mean global here, i.e. average over ALL longitudes and latitudes? It would be perfectly fine to use the regional annual average here. p. 7, l. 1: does total productivity refer to net or gross productivity? I guess net, please clarify. p. 8, l. 2: 'concentrations' p. 8, l. 2: The sentence starting here is difficult to understand. Maybe splitting into several sentences will help. p. 9, l. 12: remove 'than CTRL' p. 9, l. 20: 'macro-nutrient' p. 9, l. 21: the part '...and, productivity...' does not make sense to me. Maybe there's something missing? p. 10, l. 4: reorder sentence to '... as function of changes in summer wind stress ad curl in the central Arabian Sea.' p. 10, l. 21: use of word 'global'. I guess 'overall' or 'general' would be more appropriate here. Furthermore, the decrease is southeastward, which could be mentioned. p. 10, l. 33: replace 'suggests an' by 'demonstrates the' p. 11, l. 1: what do you mean by 'climatic precession'? I'm not familiar with the expression, so please define or use 'precession' or 'precessional index', instead. (see also next

sentence) p. 11, l. 3: the barycenter position changes with both longitude and latitude, please mention p. 11, l. 17: add 'marine sediment' before 'cores' p. 11, l. 23: 'climates" p. 11, l. 30: which latitude range are you talking about? Shouldn't a high obliquity (= high summer insolation) reduce the latitudinal temperature gradient? p. 12, l. 8: replace 'solar' by 'northern hemisphere summer' p. 12, l. 21: 'simulations" p. 12, l. 24: here is a contradiction to what you say later (l. 34). First you say that there is a mismatch to Bassinot et al., later you explain why your model matches with Bassinot et al., please clarify. p. 13, l. 13: please check use of 'globally' p. 13, l. 17-24: this part belongs to the method section p. 13, l. 27: replace 'are' by 'is' p. 13, l. 31: region up to 68°N? Please check p. 14, l. 2: I don't think that the difference between productivity and export production in particular is a major problem for model and data comparison. It is rather that there are too many unknowns on what is finally best describing the signals contained in sediment records. As it stands, the statement seems to open a completely new aspect, which demands to be further elaborated. However, since it is not the main focus of the study I'd suggest leaving it out. p. 14, l. 11-17: it seems to me that in this paragraph you write about productivity, but it is actually the export, which is explained. Please check. p. 15, l. 3: add 'or less' before 'nutrients' p. 15, l. 20: replace 'on' by 'to'. Again, this sentence seems to give a hint to problems of model-data comparison. I'd suggest adding a full paragraph on what this study contributes to a improved model-data comparison or leave it out. p. 15, l. 24: replace 'should' by 'is expected to' Figure 2, caption: 68°N? Probably this should be 28°N? Figure 8: caption: replace 'Dash' by 'Dashed' Figure 14: 'circles" or 'color of the circles'

---

## Author Comment (AC1) · 21 Jan 2017

**Response to the reviewers**

We would like to thank the editor and both reviewers for their support, comments and suggestions that helped clarify some aspects of our manuscript. We respond to all comments and questions raised by the reviewers. The reviewersqcomments are in *italic* and our answers in **bold**.

Response to reviewer#1: page 1 to 5

Response to reviewer#2: page 6 to 10

Page 11: we add an explanatory note on why we need to change figure 4, 10 and 11.

Page 12-15: we add some modified figures (due to reviewersqcomments and because of the explanation given on page11)

**Reviewer #1:**

**Recommendation**

*This paper investigates how changes in orbital parameters and ice sheets during the last glacial-interglacial cycle impact Arabian Sea productivity through changes in the monsoon intensity and spatial pattern. This is an important topic, since productivity estimates from Arabian Sea cores are usually interpreted as proxies of the monsoon intensity. Using numerical simulations with a earth system model, this study shows that the relationship between monsoon intensity and productivity is non trivial, because spatial shifts in the monsoon jet axis influence both wind stress and its curl, which both control the influence of the coastal upwelling. The paper topic is interesting, and the analyses are scientifically sound. I however suggest a couple of additional diagnostics, re-arangement of some results, additional critical discussion of the model, which should improve the clarity of the paper and make it more convincing (see general comments below).*
**We thank reviewer#1 for his very constructive comments on our work.**

**General comments**

*1. I feel that some diagnostics on the intensity of the upwelling (eg SST anomaly in the coastal box relative to the Indian Ocean average and/or vertical velocities and/or depth of the thermocline . and nutricline-) would be helpful to relate the changes in nutrients with changes in the upwelling intensity. A bit more validation of the upwelling characteristics in the present-day simulation also would not harm.*

**We agree with reviewer#1. We now evaluate the model representation of the Ekman pumping on a new panel on figure 3, on which we remove the wind intensity evaluation panel since we already evaluate the wind stress and curl (see new figure 3 at the end of the document).**
**We already have on figure 5 the changes of Ekman pumping intensity in one of the studied climate. We add the Ekman pumping changes in contour over the productivity changes of figure 7 (See new figure 7 at the end of the document)**

*2. There is no discussion of how the biases of the model for the current-day climate (e.g. its underestimated productivity) may influence the overall results of the study.*

**We started to give some explanation on the effects of the model biases on page 15 and on page 6 but we can develop the discussion on this subject. We have added a new paragraph on the model limitations to page 15.**

*3. The abstract could be improved (suggestions below). It may also be beneficial to show and discuss Fig. 14 much earlier in the paper, in order to describe the relative effects of alongshore stress and near-shore Ekman pumping in the present-day climate ahead of the past climates discussion.*

**We have taken this suggestion into account for the abstract (see below).**

We have not moved figure 14 to the evaluation section and its description in the evaluation section on p5 l35. It helps justifying that the mechanisms behind productivity changes in the Arabian Sea are represented in the model even though the amplitude of the changes in productivity and winds are underestimated.

*4. I am not sure that it is worth discussing the central Arabian Sea region (Figures 6 and 9). A reason for that is that this region is usually viewed as highly influenced by what happens in the coastal upwelling region. I hence feel that focussing on the coastal box is enough. If you want to keep this central Arabian Sea box, a diagnostic as that of figure 14, which relates the interannual variability of productivity to wind stress (here an indicator of vertical mixing) and wind stress curl would be helpful.*
**This region allows us to discuss changes in the monsoon pattern, in particular Fig. 9b but we will remove the figures and the reference to it throughout the text and just add a few words on this region in the discussion.**

*5. It may be useful to show maps of the JJAS climatological wind stress and Ekman pumping values for all the simulations, to more visually relate how the changes in low pressures over the continent relate to changes in the monsoon flow.*
**It is difficult to distinguish changes between the different climates on such maps. We do not think it would be clear enough. We believe that Figure10, which shows the link between the barycenters position and the wind stress and wind stress curl intensity is sufficient to show the link between changes in the low pressures and changes in the monsoon flow.**

**Detailed comments:**
*P1, L1: maybe ‰the current-climate Indian monsoon. . .+* **Ok**

*P1, L5-10: I think that the abstract could be clarified. Maybe mention explicitly that coastal upwelling is fuelled by a combination of alongshore stress intensity and upward Ekman pumping to the west of the jet axis. There is however strong downward Ekman pumping to the east of the jet axis, so that changes in coastal alongshore stress / curl depend both on the jet intensity and position. You can then relate changes in the intensity and position to the exact position of the low pressure over Tibet, with astronomical parameters having impact mostly on the intensity and changes in ice sheets rather influencing the jet position.*
**Ok, we have changed P1, L5-10 to: Í Locally, productivity is fueled by nutrient supply driven by Ekman dynamics. Upward transport of nutrients is modulated by a combination of alongshore wind stress intensity, which drives coastal upwelling, and by a positive wind stress curl to the west of jet axis resulting in upward Ekman pumping. On the East of the jet axis there is however a strong downward Ekman pumping due to a negative wind stress curl. Consequently, changes in coastal alongshore stress/curl depend both on the jet intensity and position. The jet position is constrained by the Indian summer monsoon pattern, which in turn is influenced by the orbital parameters and the ice sheet cover.Î**

*P1, L18: Another useful reference here is McCreary et al. (2009) (see full reference below).* **Ok**

*P1, L24: maybe indicate ‰upward+ Ekman pumping. It may be interesting to mention offshore downward pumping to the right of the jet axis.*
**Ok, we have added Í upwardÎ or Í downwardÎ in front of Í Ekman pumpingÎ throughout the text and in the Figure captions (e.g. Figure 5).**
**We have changed the text such as in P1 L24-25 : Í On the other side of the axis jet however, the negative wind stress curl generates downward Ekman pumping. Coastal upwelling and upward Ekman pumping are responsible..Î**

*P2, L5: Also mention the specific role of eddies, quoting Resplandy et al. (2011).* **We have added a sentence on the role of eddies. We will also mention it in the paragraph where we discuss the model limitations at the end of the paper.**

*P2, L9-11: Recent studies indicate that, due to changes in atmospheric stability, an increase in rainfall is not necessarily associated with an increase of the associated circulation (e.g. Held and Soden J. Clim. 2006 in the context of anthropogenic climate change).* **Ok. On P2 L11 we have added Í However, Held and Soden (2006)**

indicate that, in the context of anthropogenic climate change, an increase in rainfall is not necessarily associated with an increase of the associated circulation due to changes in atmospheric stability. This result questions the reliability of such an indicator for the monsoon intensity.Î

*P2, L24: Be more specific: change of ice sheet heights in which regions? Same comment for ice volume L27.* **We have changed it to: Î changes in ice-sheet height between glacial and interglacial climatesÎ on L24 which includes both changes in the northern and southern hemisphere and to Î changes in ice volume, especially in the Northern HemisphereÎ on L27**

*P3, L14 and following: ‰he LMDZ5A atmospheric general circulation model+ (and likewise for the other components)* **Ok**

*P4, L30-34: A recent study (Keerthi et al. 2016, see below for full ref) shows that, in winter (when the cloud cover over the Arabian Sea is low), there is a good agreement between various satellite datasets for the Northern Arabian sea, but large differences in terms of amplitude. You may hence want to add a cautionary note about uncertainties of the observational estimate.* **We have added a note about observation uncertainties**

*P5, L1: vertical mixing is more specifically due to convective overturning in presence of strong southward winds that bring dry, cold continental air.* **We have changed P4L34 P5L1 to Î In boreal winter, the mechanisms behind productivity changes are different compared to summer ones, because the winds reverse and blow from the north-east to the south-west. The presence of strong southward winds generates a convective overturning which induces vertical mixing and brings nutrients to the surface.Î**

*P5, L11: typo: ? -> ±* **yes this is a typo**
*How is productivity computed from SeaWifs?* **Productivity is computed using the VGPM algorithm on SeaWifsՅ chlorophyll data (Behrenfeld and Falkowski 1997). We will state it in the text.**

*P5, L18-19: It is also likely that the absence of eddies in the model solution contributes to a weaker offshore export than in observations (but you note it a bit later on p6). The consequences of this underestimated productivity on your results has to be thoroughly discussed at the end of the paper.* **We agree and this will be part of the added discussion of the model biases. See response to general comments #2.**

*P5, L25-35: Time series of the mean seasonal cycle of alongshore wind-stress, near-shore wind stress curl and of an indicator of the upwelling (e.g. SST) would allow a more quantitative validation of the model than the existing figures.* **We can add a figure with the seasonal variations of SST and TPP as a function of the seasonal changes in wind stress curl and stress. We can make two plots, one for the observations and one for the model outputs.**

*P6, L10: Is ‰athway‰ appropriate in this context?* **We change it to Î linksÎ**

*P6, L23-25: You should refer to fig4 to justify this diagnostic better.* **Ok**
*I am surprised by the position of the black start on figure 4e: it is on the edge of the region with SLP anomalies < -5 hPa, which is surprising for a barycentre. Is there a gridpoint with a very large negative SLP anomaly? Or maybe I‰m colorblind: you should probably highlight the -5 hPa contour on that figure.* **To clarify, we change panel e) by only plotting the SLPa-5 contour (see figure 4 at the end of this document). The SLP anomalies lower than -5 hPa extend up to 55°N and the region on which we compute the barycenter goes up to 60°N so that the barycenter is attracted to the north and its position is at the limit of the SLPa-5 region over the Tibetan Plateau.**

*P8, L5-16: I generally agree with the interpretation, but I feel that the explanation takes twists and turns. I would reorganize as follows: the shift of the Tibetan low leads to a poleward shift of the monsoon jet (5c). This leads to weaker alongshore winds in the Somalia and stronger alongshore winds in the oman upwelling (5c). But this also brings the Ekman downwelling to the right of the jet axis closer to both Oman and Somalia coasts (5e). Both factors (alongshore, and offshore curl) contribute to a Somalia upwelling reduction, while the wind stress curl change seems to overwhelm the increased alongshore winds in the Oman region, leading to overall upwelling reduction and less nutrients (5b). The reduced wind stress also reduces the mixed layer depth in the Southern Arabian Sea (5d) but I‰m not sure it‰s so relevant to show this here.* **Ok it is clearer this way and we**

**have followed the reviewer's suggestion, except we do not write "Tibetan Low" but "SLPa-5 barycenter".**

*Rather, showing SST (which characterizes the upwelling intensity) and/or vertical velocities in the model would allow to better characterize physical changes. Showing the depth of the nutricline would also be really helpful.* **We already show the Ekman pumping on panel f) of this figure. Moreover, SST may not be a good indicator of upwelling when comparing different climate states (Emeis et al 1995). We can however show the nutricline depth in contour above the mixed layer depth on panel d).**

**Emeis, K.-C., Anderson, D.M., Doose, H., Kroon, D., Schulz-Bull, D., 1995. Sea-Surface Temperatures and the History of Monsoon Upwelling in the Northwest Arabian Sea during the Last 500,000 Years. Quat. Res. 43, 355Ë361. doi:10.1006/qres.1995.1041**

*Finally, I wonder why nutrient reduction is larger in the Oman region than in the Somalia region. Is it an effect of changes in biological uptake?* **We see two hypotheses that can explain this difference in nutrient changes. The first one is a 1D explanation based on the biological uptake as you suggested. The second one is a 3D hypothesis, which implies changes in the horizontal currents. We would need to compute the nitrate balance in the area in order to identify the different contribution of processes such as advection or diffusion. But this is beyond the scope of this paper.**

*P8, L17-23: This figure is relevant for the coastal box, but is it so relevant for the offshore box, for which a lot of the productivity changes are most likely a consequences of changes in the coastal upwelling, exported offshore by the circulation. For the coastal box, it indicates that the curl change (favouring downwelling) wins over the change in coastal winds (favouring upwelling and enhanced vertical mixing). This would be confirmed by looking at the thermocline (and nutricline) depth and/or to vertical velocities, i.e. looking at physical indicators of the intensity of the upwelling. Another interest of this plot relies on the fact that productivity is decreased for all years: i.e. the change that you see is statistically significant. Finally the last interest of this plot is that it allows to evaluate the respective roles of Ekman pumping and alongshore wind stress for interannual variability: although not very clear with the current choice of color scale for panel a, there is a tendency for a weaker decrease in productivity for strong alongshore stress, as would be expected. We would also expect to see a stronger decrease in productivity for larger negative wind stress curl anomaly. If you confirm this (i.e. by performing a bilinear fit of the productivity on the stress & curl), illustrating the competition between those two effects for interannual variability will strengthen your case for explaining changes between CTL and EH. Nota: I later came to realize that you actually show this as figure 14, but it may be interesting to show it at an earlier stage in the paper.* **We remove the central area analysis here and add a few words on it to the discussion. We can then increase the panels' size so we can better see the colors and the effect of increasing wind stress on productivity. Concerning physical indicators, we believe Ekman pumping on Figure 5 and 7 already sheds light on the links between physical and biological changes.**
**We move Fig. 14 to the evaluation section so that we can discuss the relative role of wind stress and curl earlier.**

*P9, L9-10: This statement is more confusing than helpful (where exactly in the Central AS?)* **We remove the central area of the paper and thus this sentence.**

*P9, L11: Figure 8 would be usefully complemented by scatterplots between some of the variables (and the associated correlations): alongshore wind stress is strongly controlled by DTT (panels a and b); nutrients exert a (weaker) control on productivity (b and c): I would combine these scatterplot with Figure 9a (again, I don't think that figure 9b is so relevant, because wind stress is not really an upwelling driver away from the coast and productivity changes in the central Arabian Sea may not be the result of local processes).* **We remove panel b on figure 9 which concerns the central Arabian Sea. We can add on this figure some scatterplots to stress out the links between DTT and wind stress, wind stress and wind stress curl, productivity and nutrients.**

*P9, L24-25: Doing a scatterplot of alongshore wind stress vs. coastal wind stress curl would also be helpful to characterize the relations of the two parameters that control the upwelling. I would not say they are entirely independent: it seems that there is a tendency for a negative correlation.* **Ok see previous point.**

*P10, L15-20: It may be better to start by explaining the links between the low pressure position and the wind stress and curl, and then discussing the consequences on the productivity. To understand better the links*

*between the low pressure position and wind stress intensity / curl, it would be good to add a figure with maps of the JJAS wind stress (vectors) and its curl (colors), with the position of the barycentre indicated.* **Such a figure would be difficult to read since changes are small.**

*P10, L23: simply point out that the latitude of the barycentre exerts a strong control on the coastal wind stress amplitude.* **Ok**

*P10, L28: In general, in this section mention "increase of productivity with the central longitude of the Tibetan low"* **We precise Í the longitude SLPa-5 barycentreÎ to avoid any misunderstanding.**

*P11, L1-10: I'm not that familiar with those orbital parameters. Wouldn't it be simpler to directly relate the position of the low pressure to the annual (or JJAS) solar heat flux properties over the region, and then describe more qualitatively how orbital parameters control this value? Or at least remind how these parameters influence insolation and hence the low (e.g. copy the text at lines 30-31 here).* **Ok, we will add some precisions on the respective roles of the astronomical parameters on insolation here.**

*P11, L14: Âùremove only* **Ok**

*P13, L8-9: can you explain this choice?* **No simulation of the Early Holocene with the remnant Laurentide ice sheet was available at the time we performed our analysis. It is only available for the previous version of the model IPSL-CM4 (Marzin et al 2013).**
**Marzin, C., Braconnot, P., Kageyama, M., 2013. Relative impacts of insolation changes, meltwater fluxes and ice sheets on African and Asian monsoons during the Holocene. Clim. Dyn. 41, 2267Ë2286. doi:10.1007/s00382-013-1948-9**

*P13, L17: you could locate this core, e.g. on figure 7.* **Ok we add the core position on Figure 2**

*P14, L9: provide a reference in support of this statement.* **We can refer to Banakar et al. 2005.**
**Banakar, V.K.K., Oba, T., Chodankar, A.R.R., Kuramoto, T., Yamamoto, M., Minagawa, M., 2005. Monsoon related changes in sea surface productivity and water column denitrification in the Eastern Arabian Sea during the last glacial cycle. Mar. Geol. 219, 99Ë108. doi:10.1016/j.margeo.2005.05.004**

*P14, L11-17: in the model.* **Ok. We add it on l12, l14 and l15.**

*P15, L9-19: This is nice. I would move this much earlier in the paper, because it allows to explain better the relative roles of wind strees and its curl at interannual timescales before moving to applying these explanations for climates of the past.* **We move this figure in the evaluation section.**

*P15: a through discussion on the possible consequences of the present-day climate biases in the model on the results of this study is needed.* **We develop this point of the discussion. See response to general comment #2.**

**Figures:**
*Fig. 2: also draw the Northern box.* **Ok**
*Fig 3: left panels of b, c, d have some spurious horizontal lines on them.* **This is due to the model resolution. What you see is the model grid cells, thus showing the quite coarse resolution used here.**

*Some smoothing or median filter would not harm on the sat. wind stress intensity & curl.* **Ok we regrid the observations on a 1 degree grid (see Figure 3 at the end of the document).**
*On d, also avoid saturating the colorscale.* **Ok. (see Figure 3 at the end of the document)**
*Panel b: use the same spacing between vectors for the model and data.* **We remove this panel to evaluate Ekman pumping instead (see Figure 3 at the end of the document)**

*Fig 5: it would be useful to materialize the boxes that you use for integrated diagnostics on this figure.* **Ok, we add the coastal box on the productivity panel only**

*Fig 7: draw the coastal box.* **We add Ekman pumping change on Figure 7 so we won't add the coastal box otherwise it will be unreadable. (See Figure 7 at the end of the document)**

*Figure 8: describe the whiskers in the caption. I imagine it is a confidence interval: at what significance level? How is it computed? What are the small circles on this figue (and on figure 12).* **We add the following precision to the figure caption: Í The boxplots highlight the median value (bold line), the first and the third quartile (lower and higher limits of the box) and the 95% confidence interval of the median (upper and lower horizontal tics). The dots are extreme values that happened during the 100 years of simulation.Î**
**We also add this text in the legend of Figure 12.**

*Figure 10: draw the boundaries of panels b, c, d on panel a. You can re-organize the panels into 2 x 2 to save space.* **Ok to reorganize and we add the coastal box on panel a.**

*Figure 12: it would be nice to locate the core / box that are used for figure 12 on another plot, e.g. figure 7.* **Ok, we choose to add it on Fig.2**
*Clarify in the caption which panels are model results.* **We add Í** and **simulated**, b)õ **Î in the caption.**

*Figure 13: I don't find this picture very clear. I would do it upside down: start it by changes in orbital parameters / ice sheet driving changes in the Tibetan plateau low, which impact the intensity / pattern, stress / curl, the upwelling of nutrients, etc. . . Materiaize the direction of arrows connecting boxes.* **Ok, we will reverse the order and add the direction of the arrows.**

**Reviewer #2:**

*The manuscript "From Monsoon to marine productivity in the Arabian Sea: insights from glacial and interglacial climates" by P. Le Mézo et al. is an interesting model study on the driving forces of ocean productivity in the Arabian Sea. A large number of paleo climate simulations of different warm and cold climates, forced by changes in orbital parameters, greenhouse gases and northern hemisphere ice sheets is used to systematically investigate changes in productivity patterns. Against the common paradigm that there is a straightforward relationship between pure Monsoon strength (wind mixing) and marine productivity, which was often used to infer past Monsoon changes from productivity proxies, the study highlights important aspects of changes in the Monsoon pattern and potentially opposing regional trends, for example in coastal and open ocean areas. According to the model results, wind stress curl, the changes of which are mostly driven by changes in the Monsoon pattern, turns out to be an equally important factor influencing productivity. The study is well designed, straightforwardly carried out and the demonstrated results are robust and clearly illustrated in many figures. The text could benefit from a number of clarifications, however, this does not compromise the overall excellent quality of the study. I suggest publishing after some minor revisions.*
**We thank reviewer #2 for his thoughtful and useful comments that helped improve the manuscript.**

**Major comments**

*Since many readers are maybe not too familiar with the theory of wind stress curl, a short introduction into the most important mechanisms of upward nutrient transport and their interplay would probably be helpful.* **Ok we will add a more detailed description of the mechanisms in the introduction.**

*Apart from MIS3, the control simulation stands out as the most productive period for the Arabian Sea. Can the results be extrapolated to other ocean areas, for example the prominent eastern boundary upwelling regions (EBUs)? What would be the consequences for the underlying oxygen minimum zone?* **We have not looked at other regions. The recently submitted paper Bopp et al (in review) analyzes the changes in oxygen in the Indian Ocean in the LGM. The following 2 figures show the averaged changes in oxygenation between the LGM and the current day climate as simulated by the model and reconstructed in Jaccard and Galbraith 2012, between 0-1500m and 2000-5000m.**

[Figure]

**The analysis of other upwelling systems is not the scope of this study.**

**Bopp, L., Resplandy, L., Untersee, A., Le Mézo, P., Kageyama, M., in review. Ocean (de)oxygenation from the Last Glacial Maximum to the 21 st century : insights from Earth System Models. Philosophical Transactions of the Royal Society.**

*What are the model limitations? Does spatial resolution play a role? Since a rather coarse resolution was used here, what are the uncertainties and/or probable effects of parameterizations of sub gridscale processes or why*

*would all these not matter? A first tentative explanation is given on page 15, l. 6-8, but this could be further elaborated.* **We wrote about some effects of the resolution on p6 but will add a more complete paragraph on the model limitations and biases and how they affect our results and interpretation in the discussion section of the paper.**

*When describing changes in wind stress curl often the words higher or lower are used, which may cause misunderstandings. Although the relative effect of the curl tendency on upward nutrient transport is the same, it needs to be kept in mind that the change maybe associated with a change of sign of the curl or not. I׳d therefore suggest that the use of more or less negative/positive curl is a more correct wording. Please check throughout the text.* **This is a good suggestion. We have changed it throughout the text.**

**Minor points**

*p. 1, l. 6-8: This sentence is confusing. It seems that the combination of increased wind stress and more positive curl could result in a reduction of productivity. However, this is not the case (Fig. 9). I think it is more correct to state that both changes in wind stress and curl need to be taken into account and that under opposing changes one or the other may be the ultimately dominating effect.* **We change L5-10 to: ´Locally, productivity is fueled by nutrient supply driven by Ekman dynamics. Upward transport of nutrients is modulated by a combination of alongshore wind stress intensity, which drives coastal upwelling, and by a positive wind stress curl to the west of jet axis resulting in upward Ekman pumping. On the East of the jet axis there is however a strong downward Ekman pumping due to a negative wind stress curl. Consequently, changes in coastal alongshore stress/curl depend both on the jet intensity and position. The jet position is constrained by the Indian summer monsoon pattern, which in turn is influenced by the orbital parameters and the ice sheet cover.´**

*p. 1, l. 12: typo, remove ׳s׳ in ׳affects׳* **Ok**
*p. 1, l. 21 : replace ׳in India׳ by ׳over India׳* **Ok**
*p. 3, l. 15: add ׳semi-labile׳ before ׳dissolved organic carbon׳* **Ok**
*p. 3, l. 29: add ׳greenhouse׳ before ׳gas׳* **Ok**

*p. 3, l. 30: add ׳Northern Hemisphere׳ before ice sheets (at least from Fig. 1 this seems to be the case)* **We do not change it because there are some small changes in the Southern Hemisphere as well. You can mostly see it on the tip of South America on Figure1.**

*p. 4, l. 8: This sentence is confusing. Do you count MIS3 as glacial or interglacial? If glacial then you should mention that five glacial simulations were carried out. Otherwise, it is strange to see it listed among the glacial climates. Please check.* **MIS3 is counted as a glacial climate. It is indeed 5 glacial simulations and not 4. We correct that.**

*p. 4, l. 9: remove ׳or׳* **Ok**

*p. 4, l. 26: a few more information on length of simulation and starting conditions would be nice* **You can find information on the forcing parameters in Table 1. We can add information on the length of the simulation.**

*p. 4, l. 31: I guess that 68_N should be 28_N* **Yes that׳s a typo, we will change it throughout the text as it appears several times.**

*p. 4, l. 33: replace ׳sediments bulk׳ by ׳bulk sediment׳* **Ok**

*p. 4, l. 34: replace ׳changes׳ by ׳peaks׳ and ׳ones׳ by ׳maxima׳* **Ok**

*p. 4, l. 34: Does the sentence starting here refer to the winter period?* **Yes**
*If yes, then p. 5, l. 1: add ׳dashed line in׳ before ׳Fig. 2b)* **Ok**

*p. 5, l. 11: typo: NASA׳s* **Ok**

*p. 5, l. 16: ׳Arabian׳ instead of ׳Arabic׳* **Ok**

*p. 5, l. 29: what do you mean by reconstructions? Observations? Please explain.* **We meant the re-analyses, we have changed this to avoid confusion.**

*p. 6, l. 1: move 'coarse' before 'model'* **Ok**

*p. 6, l. 6 'models'* **Ok**

*p. 6, l. 20: in several places of the manuscript I'm wondering about the use of the word 'global'. Do you really mean global here, i.e. average over ALL longitudes and latitudes? It would be perfectly fine to use the regional annual average here.* **In this case we mean the average on all latitudes and longitudes.**

*p. 7, l. 1: does total productivity refer to net or gross productivity? I guess net, please clarify.* **Yes, it refers to the net productivity, we precise it in the text.**

*p. 8, l. 2: 'concentrations'* **Ok**

*p. 8, l. 2: The sentence starting here is difficult to understand. Maybe splitting into several sentences will help.* **Ok, we also remove the part on central AS as suggested by reviewer #1 (see general comment #4).**

*p. 9, l. 12: remove 'han CTRL'* **Ok**

*p. 9, l. 20: 'macro-nutrient'* **Ok**

*p. 9, l. 21: the part '..and, productivity...' does not make sense to me. Maybe there's something missing?* **We remove this part since the link between productivity and nutrients is already stated in the previous sentence.**

*p. 10, l. 4: reorder sentence to '.. as function of changes in summer wind stress ad curl in the central Arabian Sea.'* **It is clearer this way but we remove the central Arabian Sea as suggested by reviewer #1 (see general comment #4) so we also remove this sentence.**

*p. 10, l. 21: use of word 'global'. I guess 'overall' or 'general' would be more appropriate here. Furthermore, the decrease is southeastward, which could be mentioned.* **We use "overall" and add " as it moves southeastward".**

*p. 10, l. 33: replace 'suggests an' by 'demonstrates the'* **Ok**

*p. 11, l. 1: what do you mean by 'climatic precession'? I'm not familiar with the expression, so please define or use 'precession' or 'precessional index' instead. (see also next sentence)* **We only defined the climatic precession in the figure caption. We clarify that in the text p.11 l.1:**
**" On figure 11, we plotted the values of the climatic precession (defined as $e*sin(w- 180°)$ where $e$ is the eccentricity and $w-180°$ is the precession) and obliquity Å .". And we will add some more information on how the astronomical parameters affect the insolation.**

*p. 11, l. 3: the barycenter position changes with both longitude and latitude, please mention* **We change the sentence to " Climatic precession influences the SLPa-5 barycenter position both in latitude and longitude".**

*p. 11, l. 17: add 'marine sediment' before 'cores'* **Ok**

*p. 11, l. 23: 'climates+* **Ok**

*p. 11, l. 30: which latitude range are you talking about? Shouldn't a high obliquity (= high summer insolation) reduce the latitudinal temperature gradient?* **We talk about the latitudes used to compute the ΔTT value (10°N-45°N) and (25°S-10°N). In our simulations a high obliquity, increases the summer insolation in the northern hemisphere high latitudes and reduces it in the low latitudes. This higher summer insolation in the NH warms the continent and increases the tropospheric temperature gradient in boreal summer, which is characteristic of a stronger summer monsoon.**

*p. 12, l. 8: replace ‹solar› by ‹northern hemisphere summer›* **Ok**

*p. 12, l. 21: ‹simulations›+* **Ok**

*p. 12, l. 24: here is a contradiction to what you say later (l. 34). First you say that there is a mismatch to Bassinot et al., later you explain why your model matches with Bassinot et al., please clarify.*
**We mean that simulated productivity on the whole coastal area on which we made our analysis is not consistent with Bassinot et al. reconstructions. However, if we only focus on the northern part of the coastal area, our results match Bassinot et al's reconstruction for the mid-Holocene. Still they do not match the early Holocene reconstruction.**
**We have changed the formulation : Í However, even if the simulated coastal summer productivity compares quite well to the data Rostek et al. 1997 (except for the LGM, their core is in the southern part of the coastal area), it shows some discrepancies when compared to the reconstructions in Bassinot et al. 2011 (their core is in the northern part of the coastal area).Î**

*p. 13, l. 13: please check use of ‹globally›* **We change it to largely**

*p. 13, l. 17-24: this part belongs to the method section* **Ok we move it to the Method section**

*p. 13, l. 27: replace ‹are› by ‹is›* **Ok**

*p. 13, l. 31: region up to 68_N? Please check* **Yes same typo as previously. We change it to 28N.**

*p. 14, l. 2: I don't think that the difference between productivity and export production in particular is a major problem for model and data comparison. It is rather that there are too many unknowns on what is finally best describing the signals contained in sediment records. As it stands, the statement seems to open a completely new aspect, which demands to be further elaborated. However, since it is not the main focus of the study I'd suggest leaving it out.* **We have chosen to keep it here but state more clearly that it has to be viewed as a perspective of this work.**

*p. 14, l. 11-17: it seems to me that in this paragraph you write about productivity, but it is actually the export, which is explained. Please check.* **We wanted to explain primary productivity but it is true that we miss some arguments and that this description better fits the export production. We change it to make it clear.**

*p. 15, l. 3: add ‹or less› before ‹nutrients›* **Ok**

*p. 15, l. 20: replace ‹on› by ‹to›* **Ok**
*Again, this sentence seems to give a hint to problems of model data comparison. I'd suggest adding a full paragraph on what this study contributes to a improved model-data comparison or leave it out.*
**We change the sentence to: Í This study allows us to draw attention to certain points that may affect the reconstruction of past climate and productivity and hence the comparison between model and data.Î**

*p. 15, l. 24: replace ‹should› by ‹is expected to›* **Ok**

**Figures:**
*Figure 2, caption: 68_N? Probably this should be 28_N?* **Yes**
*Figure 8: caption:replace ‹Dash› by ‹Dashed›* **Ok**
*Figure 14: ‹circles›+or ‹color of the circles›* **Ok**

**NOTE:**

We realized that the computing of the barycenter was erroneous: in the submitted version, the barycenter was computed by taking the SLP anomalies relative to the regional mean (20W-150E,30S-60N) and not to the global mean as we intended. We corrected that. This correction made us change figures 4, 10 and 11. The barycenters move slightly north, especially in some glacial climates, but this does not change our conclusions. The new figures are added at the end of this document.

We added a few word in section 3.2.2 p10 l.33:

**%This latitudinal movement of the SLPa-5 barycentre with the monsoon strength seems also to be dependent on the glacial or interglacial state of the simulation. Indeed, in glacial simulations the SLPa-5 barycentre is north of the CTRL barycentre even if the monsoons are less intense (LGM, MIS4D and MIS4M), however the other glacial simulations with higher summer monsoon intensity (MIS3 and MIS4F) have their barycentre located north of these glacial simulations (Fig. 10). Similarly, in inter-glacial climates, the Holocene simulations have a barycentre north of the CTRL one (Fig. 10).Î**

And we change a sentence on p.12 l.9-10
**%The changes in the ITCZ position highlighted by Fleitmann et al. (2007) are consistent with our results, especially with the changes in the position of the SLPa-5 barycentre in latitude given a similar background state: glacial or interglacial** (Fig.10).+
instead of %The changes in the ITCZ position highlighted by Fleitmann et al. (2007) are consistent with our results, especially with 10 the changes in the position of the SLPa-5 barycentre in latitude (Fig.10).+

We add a missing %**on**+on p.13 l.9
We remove an extra %**climate**+on p.15, l.21

**Figure 3 modified. We remove wind intensity and add Ekman pumping.**

[Figure]

**We modify the evaluation section p5 L10-35 accordingly:**

Í We use satellite products from remote sensing by NASA's Sea-viewing Field-of-view Sensor (SeaWIFS) during the period 1998-2005 for monthly productivity (Lévy et al., 2007) and the NOAA Multiple-Satellite 1995-2005 climatological cycle **for wind stress intensity** (Zhang, 2006). We compute the observed and modelled wind stress curl intensity **and Ekman pumping** from the wind stress data and model output, respectively.

Figure 3a shows that simulated boreal summer productivity integrated over the whole water column is underestimated relative to the reconstructed boreal summer productivity, especially in the regions of upwelling, along the coast of the Arabic Peninsula and Somalia. The spatial correlation coefficient, R, between the observed and simulated productivity is 0.44. ~~Underestimation of productivity is first caused by an underestimated wind intensity, which affects the extent and intensity of the coastal upwelling and the supply of nutrients to the surface layer. The boreal summer wind patterns, which are characteristic of the boreal summer monsoon system, are better represented than productivity with a correlation of 0.86. Sperber et al. (2013) studied the representation of the Asian summer monsoon in the CMIP5 models, which comprises the IPSL model. They showed that the monsoon was better represented in the CMIP5 models compared to the CMIP3 models, especially the monsoonal winds. We can however note that the alongshore winds in the western Arabian Sea have a more northerly orientation in the CTRL simulation than in the observations, which can affect the dynamical processes in the region (Fig. 3b).~~

In the Arabian Sea, summer productivity is affected by the winds through different mechanisms implying the wind stress and the wind stress curl (Anderson et al., 1992). The strong winds along the Arabian coast, called the Somali Jet, generate a positive wind stress which increases Ekman transport off the coast. The water that leaves the coastal area is being replaced by subsurface water: this is the coastal upwelling.  **T**he CTRL simulation wind stress intensity is **characteristic of the boreal summer monsoon system and better represented than productivity. Sperber et al. (2013) studied the representation of the**

Asian summer monsoon in the CMIP5 models, which comprises the IPSL model. They showed that the monsoon was better represented in the CMIP5 models compared to the CMIP3 models, especially the monsoonal winds. **The alongshore wind stress intensity is however** underestimated compared to the **observations**: the maximum wind stress intensity is lower and it does not extend far north in the Arabian Sea as the **observed** wind stress (Fig.3**b**). The wind stress orientation is also more zonal in the simulation than in the observations (Fig. 3b, e). Figure 3 **c** represents the wind stress curl, computed from the wind stress, in the simulation and in the observations. [..] . **Finally, we evaluate the Ekman pumping intensity on figure 3d. Ekman pumping intensity pattern resembles the wind stress curl pattern has it drives the Ekman dynamics. We see close to the Arabian coast a positive Ekman pumping which is characteristic of the coastal upwelling generated by the positive wind stress and wind stress curl (Fig.3b, c, d). On the other side of the axis jet, in the center of the Arabian Sea, wind stress curl is negative which creates downwelling characterized by a negative Ekman pumping (Fig.3d). The model and the data show similar pattern but slightly different intensities (Fig.3d). The difference of Ekman pumping spatial distributions between model and data arise from the differences in the wind stress curl intensity which in turn, are the results of differences in the wind patterns (Fig.3).**

**Figure 4 :**

[Figure]

**Figure 4: Panels e and f modified (we only put the SLPa-5 on panel e and change the stars position)**

[Figure]

**Figure 7 modified. We add Ekman pumping changes and core location (yellow circle)**

[Figure]

**Figure 10 modified. New barycenter' position**

[Figure]

**Figure 11 modified. New barycenter' position**

---

## Author Response (AR1)

**Response to the reviewers**

We would like to thank the editor and both reviewers for their support, comments and suggestions that helped clarify some aspects of our manuscript. We respond to all comments and questions raised by the reviewers. The reviewers' comments are in *italic* and our answers in **bold**.

**Reviewer #1:**

**Recommendation**

*This paper investigates how changes in orbital parameters and ice sheets during the last glacial-interglacial cycle impact Arabian Sea productivity through changes in the monsoon intensity and spatial pattern. This is an important topic, since productivity estimates from Arabian Sea cores are usually interpreted as proxies of the monsoon intensity. Using numerical simulations with a earth system model, this study shows that the relationship between monsoon intensity and productivity is non trivial, because spatial shifts in the monsoon jet axis influence both wind stress and its curl, which both control the influence of the coastal upwelling. The paper topic is interesting, and the analyses are scientifically sound. I however suggest a couple of additional diagnostics, re-arangement of some results, additional critical discussion of the model, which should improve the clarity of the paper and make it more convincing (see general comments below).*

**We thank reviewer#1 for his very constructive comments on our work.**

**General comments**

1. *I feel that some diagnostics on the intensity of the upwelling (eg SST anomaly in the coastal box relative to the Indian Ocean average and/or vertical velocities and/or depth of the thermocline . and nutricline-) would be helpful to relate the changes in nutrients with changes in the upwelling intensity. A bit more validation of the upwelling characteristics in the present-day simulation also would not harm.*

**We add an evaluation of the SST anomalies in boreal summer on a new panel on figure 3.**

2. *There is no discussion of how the biases of the model for the current-day climate (e.g. its underestimated productivity) may influence the overall results of the study.*

**We started to give some explanation on the effects of the model biases on page 15 and on page 6 but we can develop the discussion on this subject.**

**We have added a new paragraph on the model limitations to page 15 L10-21.** Í We need to keep in mind that the model's coarse resolution does not allow for a very precise representation of the region dynamics. This may have altered the relative weight of the processes related to wind stress and wind stress curl and can explain why the astronomical signal is weak in the productivity changes. **Arabian Sea productivity in the CTRL simulation shows quite large differences with data, especially for the high coastal productivity extension in the western Arabian Sea (Fig. 3a). This can results from the model coarse resolution which prevents the representation of meso-scale processes such as eddies. These fine-scale processes are shown to be of importance for the coupling between biology and physics (Resplandy et al., 2011). Moreover, these meso-scale processes contribute strongly to the export of nutrients offshore and thus can explain why our high productivity area is more restricted to the coast than in the observations. From our set of simulations we cannot assess the effect of the underestimation of productivity on our results. Our simulations may underestimate productivity levels and variations but since most of the main physical processes are represented we can draw conclusions on the link between these processes and productivity and their changes through time. A way to overcome these limitations and to quantify their effects would be to work with several other models and to analyse the coupling between biology and physics in those different models.Î**

3. *The abstract could be improved (suggestions below). It may also be beneficial to show and discuss Fig. 14 much earlier in the paper, in order to describe the relative effects of alongshore stress and near-shore Ekman pumping in the present-day climate ahead of the past climates discussion.*

**We have taken this suggestion into account for the abstract (see below).**

**We have moved figure 14 to the evaluation section and its description in the evaluation section on p6 L26-34. It helps justifying that the mechanisms behind productivity changes in the Arabian Sea are represented in the model even though the amplitude of the changes in productivity and winds are underestimated.**

*4. I am not sure that it is worth discussing the central Arabian Sea region (Figures 6 and 9). A reason for that is that this region is usually viewed as highly influenced by what happens in the coastal upwelling region. I hence feel that focussing on the coastal box is enough. If you want to keep this central Arabian Sea box, a diagnostic as that of figure 14, which relates the interannual variability of productivity to wind stress (here an indicator of vertical mixing) and wind stress curl would be helpful.*
**This region allows us to discuss changes in the monsoon pattern, in particular Fig. 9b but we remove the figures and the reference to it throughout the text.**

*5. It may be useful to show maps of the JJAS climatological wind stress and Ekman pumping values for all the simulations, to more visually relate how the changes in low pressures over the continent relate to changes in the monsoon flow.*
**It is difficult to distinguish changes between the different climates on such maps. We do not think it would be clear enough. We believe that Figure11 (old figure10), which shows the link between the barycenterS position and the wind stress and wind stress curl intensity is sufficient to show the link between changes in the low pressures and changes in the monsoon flow.**

**Detailed comments:**

*P1, L1: maybe ‰the current-climate Indian monsoon. . .+* **Ok**

*P1, L5-10: I think that the abstract could be clarified. Maybe mention explicitly that coastal upwelling is fuelled by a combination of alongshore stress intensity and upward Ekman pumping to the west of the jet axis. There is however strong downward Ekman pumping to the east of the jet axis, so that changes in coastal alongshore stress / curl depend both on the jet intensity and position. You can then relate changes in the intensity and position to the exact position of the low pressure over Tibet, with astronomical parameters having impact mostly on the intensity and changes in ice sheets rather influencing the jet position.*
**Ok, we have changed P1, L5-10 to: Í Locally, productivity is fueled by nutrient supply driven by Ekman dynamics. Upward transport of nutrients is modulated by a combination of alongshore wind stress intensity, which drives coastal upwelling, and by a positive wind stress curl to the west of jet axis resulting in upward Ekman pumping. On the East of the jet axis there is however a strong downward Ekman pumping due to a negative wind stress curl. Consequently, changes in coastal alongshore stress/curl depend both on the jet intensity and position. The jet position is constrained by the Indian summer monsoon pattern, which in turn is influenced by the orbital parameters and the ice sheet cover.Î**

*P1, L18: Another useful reference here is McCreary et al. (2009) (see full reference below).* **Ok**

*P1, L24: maybe indicate ‰upward+ Ekman pumping. It may be interesting to mention offshore downward pumping to the right of the jet axis.*
**Ok, we have added Í upwardÎ or Í downwardÎ in front of Í Ekman pumpingÎ throughout the text and in the Figure captions (e.g. Figure 6).**
**We have changed the text such as in P1 L23-24 P2 L1-6:**
**Í In addition the wind's tendency to turn on itself in the horizontal plane, quantified by the wind stress curl, also drives upward and downward water transport (**Marshall and Plumb, 2008**). Between the axis of the jet and the western coast, the wind stress is cyclonic and the wind stress curl is positive, it drives a divergent flow that causes upward Ekman pumping (**Murtugudde et al., 2007; Barber et al., 2001; Anderson et al., 1992; Findlater, 1969). **On the other side of the axis jet however, the wind stress is anticyclonic, the wind stress curl is therefore negative and drives a convergent flow and downward Ekman pumping. Coastal upwelling and upward Ekman pumping are responsible for increased productivity in the western coastal Arabian** due to a higher supply of nutrients to the surface layer (Anderson et al., 1992; Anderson and Prell, 1992)."

*P2, L5: Also mention the specific role of eddies, quoting Resplandy et al. (2011).*
**We change P2 L11-12** : "and increase productivity in those regions ( Wiggert et al., 2005; Prasanna Kumar et al., 2001; Lee et al., 2000). **At the mesoscale, filaments contribute to the lateral advection of nutrients form the coast to the central Arabian Sea (Resplandy et al., 2011)."**

*P2, L9-11: Recent studies indicate that, due to changes in atmospheric stability, an increase in rainfall is not necessarily associated with an increase of the associated circulation (e.g. Held and Soden J. Clim. 2006 in the context of anthropogenic climate change).* **Ok. On P2 L14 we have added "However, Held and Soden (2006) indicate that, in the context of anthropogenic climate change, an increase in rainfall is not necessarily associated with an increase of the associated circulation due to changes in atmospheric stability. This result questions the reliability of such an indicator for the monsoon intensity."**

*P2, L24: Be more specific: change of ice sheet heights in which regions? Same comment for ice volume L27.* **We have changed it to: "changes in ice-sheet height between glacial and interglacial climates" on L33 which includes both changes in the northern and southern hemisphere and to "changes in ice volume, especially in the Northern Hemisphere" on L35**

*P3, L14 and following: "the LMDZ5A atmospheric general circulation model" (and likewise for the other components)* **Ok**

*P4, L30-34: A recent study (Keerthi et al. 2016, see below for full ref) shows that, in winter (when the cloud cover over the Arabian Sea is low), there is a good agreement between various satellite datasets for the Northern Arabian sea, but large differences in terms of amplitude. You may hence want to add a cautionary note about uncertainties of the observational estimate.* **We do not see in Keerthi et al 2016 a discussion on satellite observations discrepancies concerning chlorophyll estimates.**

*P5, L1: vertical mixing is more specifically due to convective overturning in presence of strong southward winds that bring dry, cold continental air.* **We have changed P5L8-10 to "In boreal winter, the mechanisms behind productivity changes are different compared to summer ones, because the winds reverse and blow from the north-east to the south-west. The presence of strong southward winds generates a convective overturning which induces vertical mixing and brings nutrients to the surface."**

*P5, L11: typo: ? -> +* **yes this is a typo**
*How is productivity computed from SeaWifs?* **Productivity is computed using the VGPM algorithm on SeaWifs chlorophyll data (Behrenfeld and Falkowski 1997).**
**We add it in the text P5-L20-21 : "during the period 1998-2005 processed with the VGPM algorithm (Behrenfeld and Falkowski, 1997) to obtain monthly productivity"**

*P5, L18-19: It is also likely that the absence of eddies in the model solution contributes to a weaker offshore export than in observations (but you note it a bit later on p6). The consequences of this underestimated productivity on your results has to be thoroughly discussed at the end of the paper.* **We add it in the discussion of the model biases. See response to general comments #2.**

*P5, L25-35: Time series of the mean seasonal cycle of alongshore wind-stress, near-shore wind stress curl and of an indicator of the upwelling (e.g. SST) would allow a more quantitative validation of the model than the existing figures.* **We only focus on the boreal summer season and don't feel the need to evaluate the simulation in winter as well. Moreover, the use of such maps allows us to evaluate the distribution patterns of the different variables.**

*P6, L10: Is "pathway" appropriate in this context?* **We change it to "links"**

*P6, L23-25: You should refer to fig4 to justify this diagnostic better.* **Ok**
*I am surprised by the position of the black start on figure 4e: it is on the edge of the region with SLP anomalies < -5 hPa, which is surprising for a barycentre. Is there a gridpoint with a very large negative SLP anomaly? Or maybe I'm colorblind: you should probably highlight the -5 hPa contour on that figure.* **To clarify, we change panel e) by only plotting the SLPa-5 contour. The SLP anomalies lower than -5 hPa extend up to 55°N**

and the region on which we compute the barycenter goes up to 60°N so that the barycenter is attracted to the north and its position is at the limit of the SLPa-5 region over the Tibetan Plateau.

*P8, L5-16: I generally agree with the interpretation, but I feel that the explanation takes twists and turns. I would reorganize as follows: the shift of the Tibetan low leads to a poleward shift of the monsoon jet (5c). This leads to weaker alongshore winds in the Somalia and stronger alongshore winds in the oman upwelling (5c). But this also brings the Ekman downwelling to the right of the jet axis closer to both Oman and Somalia coasts (5e). Both factors (alongshore, and offshore curl) contribute to a Somalia upwelling reduction, while the wind stress curl change seems to overwhelm the increased alongshore winds in the Oman region, leading to overall upwelling reduction and less nutrients (5b). The reduced wind stress also reduces the mixed layer depth in the Southern Arabian Sea (5d) but I´m not sure it´s so relevant to show this here.* **Ok it is clearer this way and we have followed the reviewer´s suggestion, except we do not write Í Tibetan LowÎ but Í SLPa-5 barycenterÎ.**

*Rather, showing SST (which characterizes the upwelling intensity) and/or vertical velocities in the model would allow to better characterize physical changes. Showing the depth of the nutricline would also be really helpful.* **We already show the Ekman pumping on panel f) of this figure. Moreover, SST may not be a good indicator of upwelling when comparing different climate states (Emeis et al 1995).**

**Emeis, K.-C., Anderson, D.M., Doose, H., Kroon, D., Schulz-Bull, D., 1995. Sea-Surface Temperatures and the History of Monsoon Upwelling in the Northwest Arabian Sea during the Last 500,000 Years. Quat. Res. 43, 355Ě361. doi:10.1006/qres.1995.1041**

*Finally, I wonder why nutrient reduction is larger in the Oman region than in the Somalia region. Is it an effect of changes in biological uptake?* **We see two hypotheses that can explain this difference in nutrient changes. The first one is a 1D explanation based on the biological uptake as you suggested. The second one is a 3D hypothesis, which implies changes in the horizontal currents. We would need to compute the nitrate balance in the area in order to identify the different contribution of processes such as advection or diffusion. But this is beyond the scope of this paper.**

*P8, L17-23: This figure is relevant for the coastal box, but is it so relevant for the offshore box, for which a lot of the productivity changes are most likely a consequences of changes in the coastal upwelling, exported offshore by the circulation. For the coastal box, it indicates that the curl change (favouring downwelling) wins over the change in coastal winds (favouring upwelling and enhanced vertical mixing). This would be confirmed by looking at the thermocline (and nutricline) depth and/or to vertical velocities, i.e. looking at physical indicators of the intensity of the upwelling. Another interest of this plot relies on the fact that productivity is decreased for all years: i.e. the change that you see is statistically significant. Finally the last interest of this plot is that it allows to evaluate the respective roles of Ekman pumping and alongshore wind stress for interannual variability: although not very clear with the current choice of color scale for panel a, there is a tendency for a weaker decrease in productivity for strong alongshore stress, as would be expected. We would also expect to see a stronger decrease in productivity for larger negative wind stress curl anomaly. If you confirm this (i.e. by performing a bilinear fit of the productivity on the stress & curl), illustrating the competition between those two effects for interannual variability will strengthen your case for explaining changes between CTL and EH. Nota: I later came to realize that you actually show this as figure 14, but it may be interesting to show it at an earlier stage in the paper.* **We remove the central area analysis here and add a few words on it to the discussion. We can then increase the panels´ size so we can better see the colors and the effect of increasing wind stress on productivity. Concerning physical indicators, we believe Ekman pumping on Figure 6 (old figure 5) and 8 (old figure 7) already sheds light on the links between physical and biological changes.**
**We move Fig. 14 to the evaluation section (it becomes figure 4) so that we can discuss the relative role of wind stress and curl earlier.**

*P9, L9-10: This statement is more confusing than helpful (where exactly in the Central AS?)* **We remove the central area of the paper and thus this sentence.**

*P9, L11: Figure 8 would be usefully complemented by scatterplots between some of the variables (and the associated correlations): alongshore wind stress is strongly controlled by DTT (panels a and b); nutrients exert a (weaker) control on productivity (b and c): I would combine these scatterplot with Figure 9a (again, I don´t think that figure 9b is so relevant, because wind stress is not really an upwelling driver away from the coast and productivity changes in the central Arabian Sea may not be the result of local processes).*

**We remove panel b on figure 9 which concerns the central Arabian Sea.**
**We think that the scatterplots won't add more information than there is already in figure 9 (old figure 8) since there is no one-to-one relationship between most of the variables. It is quite clear when looking at the boxplots that DTT and wind stress are related and that PP and NO3 are linked as well.**

*P9, L24-25: Doing a scatterplot of alongshore wind stress vs. coastal wind stress curl would also be helpful to characterize the relations of the two parameters that control the upwelling. I would not say they are entirely independent: it seems that there is a tendency for a negative correlation.* **We already have plotted this relationship between stress and curl on Figure 10 (old figure 9) and on figure 4 (old figure 14).**

*P10, L15-20: It may be better to start by explaining the links between the low pressure position and the wind stress and curl, and then discussing the consequences on the productivity. To understand better the links between the low pressure position and wind stress intensity / curl, it would be good to add a figure with maps of the JJAS wind stress (vectors) and its curl (colors), with the position of the barycentre indicated.* **Such a figure would be difficult to read since changes are small.**

*P10, L23: simply point out that the latitude of the barycentre exerts a strong control on the coastal wind stress amplitude.* **Ok**

*P10, L28: In general, in this section mention "increase of productivity with the central longitude of the Tibetan low".* **We precise Í the longitude SLPa-5 barycentreÎ to avoid any misunderstanding.**

*P11, L1-10: I'm not that familiar with those orbital parameters. Wouldn't it be simpler to directly relate the position of the low pressure to the annual (or JJAS) solar heat flux properties over the region, and then describe more qualitatively how orbital parameters control this value? Or at least remind how these parameters influence insolation and hence the low (e.g. copy the text at lines 30-31 here).* **Ok, we will add some precisions on the respective roles of the astronomical parameters on insolation here. P11 L8-11 Í On figure 12, we plotted the values of the climatic precession (defined as e\*sin($w - 180°$) with e the eccentricity and $w$ the precession), which modulates the northern hemisphere insolation, and obliquity, which controls the temperature contrast between the high and low latitudes, relative to the SLPa-5 barycentre position, in order to analyse the relationship between the astronomical parameters and the SLPa-5 barycentre's position.**"

*P11, L14: «remove only* **Ok**

*P13, L8-9: can you explain this choice?* **No simulation of the Early Holocene with the remnant Laurentide ice sheet was available at the time we performed our analysis. It is only available for the previous version of the model IPSL-CM4 (Marzin et al 2013).**
**Marzin, C., Braconnot, P., Kageyama, M., 2013. Relative impacts of insolation changes, meltwater fluxes and ice sheets on African and Asian monsoons during the Holocene. Clim. Dyn. 41, 2267Ë2286. doi:10.1007/s00382-013-1948-9**
**We add P13 L12 Í The simulation with this model version was not available at the time of our analyses.Î**

*P13, L17: you could locate this core, e.g. on figure 7.* **Ok we add the core position on Figure 2 and Figure 8**

*P14, L9: provide a reference in support of this statement.* **We can refer to Banakar et al. 2005.**
**Banakar, V.K.K., Oba, T., Chodankar, A.R.R., Kuramoto, T., Yamamoto, M., Minagawa, M., 2005. Monsoon related changes in sea surface productivity and water column denitrification in the Eastern Arabian Sea during the last glacial cycle. Mar. Geol. 219, 99Ë108. doi:10.1016/j.margeo.2005.05.004**

*P14, L11-17: in the model.* **Ok. We add it on several lines.**

*P15, L9-19: This is nice. I would move this much earlier in the paper, because it allows to explain better the relative roles of wind strees and its curl at interannual timescales before moving to applying these explanations for climates of the past.* **We move this figure in the evaluation section.**

*P15: a through discussion on the possible consequences of the present-day climate biases in the model on the results of this study is needed.* **We develop this point of the discussion. See response to general comment #2.**

**Figures:**

*Fig. 2: also draw the Northern box.* **Ok**
*Fig 3: left panels of b, c, d have some spurious horizontal lines on them.* **This is due to the model resolution. What you see is the model grid cells, thus showing the quite coarse resolution used here.**

*Some smoothing or median filter would not harm on the sat. wind stress intensity & curl.* **Ok we regrid the observations on a 1 degree grid**
*On d, also avoid saturating the colorscale.* **Ok.**
*Panel b: use the same spacing between vectors for the model and data.* **Ok**

*Fig 5: it would be useful to materialize the boxes that you use for integrated diagnostics on this figure.* **Ok, we add the coastal box on the productivity panel only**

*Fig 7: draw the coastal box.* **We add Ekman pumping change on Figure 8 (old figure 7) so we won't add the coastal box otherwise it will be unreadable.**

*Figure 8: describe the whiskers in the caption. I imagine it is a confidence interval: at what significance level? How is it computed? What are the small circles on this figue (and on figure 12).* **We add the following precision to the figure caption: Í The boxplots highlight the median value (bold line), the first and the third quartile (lower and higher limits of the box) and the 95% confidence interval of the median (upper and lower horizontal tics). The dots are extreme values that happened during the 100 years of simulation.Î**
**We also add this text in the legend of Figure 13.**

*Figure 10: draw the boundaries of panels b, c, d on panel a. You can re-organize the panels into 2 x 2 to save space.* **Ok to reorganize and we add the boundaries on panel a.**

*Figure 12: it would be nice to locate the core / box that are used for figure 12 on another plot, e.g. figure 7.* **Ok**
*Clarify in the caption which panels are model results.* **We add Í and simulated, b)õ Î in the caption.**

*Figure 13: I don't find this picture very clear. I would do it upside down: start it by changes in orbital parameters / ice sheet driving changes in the Tibetan plateau low, which impact the intensity / pattern, stress / curl, the upwelling of nutrients, etc. . . Materiaize the direction of arrows connecting boxes.* **Ok, we reversed the order and add the direction of the arrows.**

**Reviewer #2:**

*The manuscript ₫From Monsoon to marine productivity in the Arabian Sea: insights from glacial and interglacial climatesₒby P. Le Mézo et al. is an interesting model study on the driving forces of ocean productivity in the Arabian Sea. A large number of paleo climate simulations of different warm and cold climates, forced by changes in orbital parameters, greenhouse gases and northern hemisphere ice sheets is used to systematically investigate changes in productivity patterns. Against the common paradigm that there is a straightforward relationship between pure Monsoon strength (wind mixing) and marine productivity, which was often used to infer past Monsoon changes from productivity proxies, the study highlights important aspects of changes in the Monsoon pattern and potentially opposing regional trends, for example in coastal and open ocean areas. According to the model results, wind stress curl, the changes of which are mostly driven by changes in the Monsoon pattern, turns out to be an equally important factor influencing productivity. The study is well designed, straightforwardly carried out and the demonstrated results are robust and clearly illustrated in many figures. The text could benefit from a number of clarifications, however, this does not compromise the overall excellent quality of the study. I suggest publishing after some minor revisions.*

**We thank reviewerr#2 for his thoughtful and useful comments that helped improve the manuscript.**

**Major comments**

*Since many readers are maybe not too familiar with the theory of wind stress curl, a short introduction into the most important mechanisms of upward nutrient transport and their interplay would probably be helpful.* **Ok we add a more detailed description of the mechanisms in the introduction. P1 L23-24 P2 L1-6:**

Í **In addition the wind's tendency to turn on itself in the horizontal plane, quantified by the wind stress curl, also drives upward and downward water transport (**Marshall and Plumb, 2008). **Between the axis of the jet and the western coast, the wind stress is cyclonic and the wind stress curl is positive, it drives a divergent flow that causes upward Ekman pumping (**Murtugudde et al., 2007; Barber et al., 2001; Anderson et al., 1992; Findlater, 1969). **On the other side of the axis jet however, the wind stress is anticyclonic, the wind stress curl is therefore negative and drives a convergent flow and downward Ekman pumping. Coastal upwelling and upward Ekman pumping are responsible for increased productivity in the western coastal Arabian** due to a higher supply of nutrients to the surface layer (Anderson et al., 1992; Anderson and Prell, 1992)."

*Apart from MIS3, the control simulation stands out as the most productive period for the Arabian Sea. Can the results be extrapolated to other ocean areas, for example the prominent eastern boundary upwelling regions (EBUs)? What would be the consequences for the underlying oxygen minimum zone?* **We have not looked at other regions. The recently submitted paper Bopp et al (in review) analyzes the changes in oxygen in the Indian Ocean in the LGM. The following 2 figures show the averaged changes in oxygenation between the LGM and the current day climate as simulated by the model and reconstructed in Jaccard and Galbraith 2012, between 0-1500m and 2000-5000m.**

[Figure]

**The analysis of other upwelling systems is not the scope of this study.**

**Bopp, L., Resplandy, L., Untersee, A., Le Mézo, P., Kageyama, M., in review. Ocean (de)oxygenation from the Last Glacial Maximum to the 21 st century : insights from Earth System Models. Philosophical Transactions of the Royal Society.**

*What are the model limitations? Does spatial resolution play a role? Since a rather coarse resolution was used here, what are the uncertainties and/or probable effects of parameterizations of sub gridscale processes or why would all these not matter? A first tentative explanation is given on page 15, l. 6-8, but this could be further elaborated.* **We wrote about some effects of the resolution on p6 but add a more complete paragraph on the model limitations and biases and how they affect our results and interpretation in the discussion section of the paper. P15 L10-21.** Í We need to keep in mind that the model's coarse resolution does not allow for a very precise representation of the region dynamics. This may have altered the relative weight of the processes related to wind stress and wind stress curl and can explain why the astronomical signal is weak in the productivity changes. **Arabian Sea productivity in the CTRL simulation shows quite large differences with data, especially for the high coastal productivity extension in the western Arabian Sea (Fig.3a). This can results from the model coarse resolution which prevents the representation of meso-scale processes such as eddies. These fine-scale processes are shown to be of importance for the coupling between biology and physics (Resplandy et al., 2011). Moreover, these meso-scale processes contribute strongly to the export of nutrients offshore and thus can explain why our high productivity area is more restricted to the coast than in the observations. From our set of simulations we cannot assess the effect of the underestimation of productivity on our results. Our simulations may underestimate productivity levels and variations but since most of the main physical processes are represented we can draw conclusions on the link between these processes and productivity and their changes through time. A way to overcome these limitations and to quantify their effects would be to work with several other models and to analyse the coupling between biology and physics in those different models.**Î

*When describing changes in wind stress curl often the words higher or lower are used, which may cause misunderstandings. Although the relative effect of the curl tendency on upward nutrient transport is the same, it needs to be kept in mind that the change maybe associated with a change of sign of the curl or not. I therefore suggest that the use of more or less negative/positive curl is a more correct wording. Please check throughout the text.* **This is a good suggestion. We have changed it throughout the text.**

**Minor points**

*p. 1, l. 6-8: This sentence is confusing. It seems that the combination of increased wind stress and more positive curl could result in a reduction of productivity. However, this is not the case (Fig. 9). I think it is more correct to state that both changes in wind stress and curl need to be taken into account and that under opposing changes one or the other may be the ultimately dominating effect.* **We change L5-10 to: Í Locally, productivity is fueled by nutrient supply driven by Ekman dynamics. Upward transport of nutrients is modulated by a combination of alongshore wind stress intensity, which drives coastal upwelling, and by a positive wind stress curl to the west of jet axis resulting in upward Ekman pumping. On the East of the jet axis there is however a strong downward Ekman pumping due to a negative wind stress curl. Consequently, changes in coastal alongshore stress/curl depend both on the jet intensity and position. The jet position is constrained by the Indian summer monsoon pattern, which in turn is influenced by the orbital parameters and the ice sheet cover.**Î

*p. 1, l. 12: typo, remove 'in' affects* **Ok**
*p. 1, l. 21 : replace 'in India' by 'over India'* **Ok**
*p. 3, l. 15: add 'semi-labile' before 'dissolved organic carbon'* **Ok**
*p. 3, l. 29: add 'greenhouse' before 'gas'* **Ok**

*p. 3, l. 30: add 'Northern Hemisphere' before ice sheets (at least from Fig. 1 this seems to be the case)* **We do not change it because there are some small changes in the Southern Hemisphere as well. You can mostly see it on the tip of South America on Figure1.**

*p. 4, l. 8: This sentence is confusing. Do you count MIS3 as glacial or interglacial? If glacial then you should mention that five glacial simulations were carried out. Otherwise, it is strange to see it listed among the glacial climates. Please check.* **MIS3 is counted as a glacial climate. It is indeed 5 glacial simulations and not 4. We correct that.**

*p. 4, l. 9: remove 'for'* **Ok**

*p. 4, l. 26: a few more information on length of simulation and starting conditions would be nice* **You can find information on the forcing parameters in Table 1. We can add information on the length of the simulation.**

*p. 4, l. 31: I guess that 68_N should be 28_N* **Yes that's a typo, we will change it throughout the text as it appears several times.**

*p. 4, l. 33: replace 'sediments bulk' by 'bulk sediment'* **Ok**

*p. 4, l. 34: replace 'changes' by 'peaks' and 'ones' by 'maxima'* **Ok**

*p. 4, l. 34: Does the sentence starting here refer to the winter period?* **Yes**
*If yes, then p. 5, l. 1: add 'dashed line in' before '(Fig. 2b)* **Ok**

*p. 5, l. 11: typo: NASA's* **Ok**

*p. 5, l. 16: 'Arabian' instead of 'Arabic'* **Ok**

*p. 5, l. 29: what do you mean by reconstructions? Observations? Please explain.* **We meant the re-analyses, we have changed this to avoid confusion.**

*p. 6, l. 1: move 'coarse' before 'model'* **Ok**

*p. 6, l. 6 'models'* **Ok**

*p. 6, l. 20: in several places of the manuscript I'm wondering about the use of the word 'global' Do you really mean global here, i.e. average over ALL longitudes and latitudes? It would be perfectly fine to use the regional annual average here.* **In this case we mean the average on all latitudes and longitudes.**

*p. 7, l. 1: does total productivity refer to net or gross productivity? I guess net, please clarify.* **Yes, it refers to the net productivity, we precise it in the text.**

*p. 8, l. 2: 'concentrations'* **Ok**

*p. 8, l. 2: The sentence starting here is difficult to understand. Maybe splitting into several sentences will help.* **Ok, we also remove the part on central AS as suggested by reviewer #1 (see general comment #4).**

*p. 9, l. 12: remove 'than CTRL'* **Ok**

*p. 9, l. 20: 'macro-nutrient'* **Ok**

*p. 9, l. 21: the part '..and, productivity...' does not make sense to me. Maybe there's something missing?* **We remove this part since the link between productivity and nutrients is already stated in the previous sentence.**

*p. 10, l. 4: reorder sentence to '.. as function of changes in summer wind stress ad curl in the central Arabian Sea.'* **It is clearer this way but we remove the central Arabian Sea as suggested by reviewer #1 (see general comment #4) so we also remove this sentence.**

*p. 10, l. 21: use of word 'global' I guess 'overall' or 'general' would be more appropriate here. Furthermore, the decrease is southeastward, which could be mentioned.* **We use "overall" and add "as it moves southeastward"**

*p. 10, l. 33: replace 'suggests an' by 'demonstrates the'* **Ok**

*p. 11, l. 1: what do you mean by 'climatic precession'? I'm not familiar with the expression, so please define or use 'precession' or 'precessional index' instead. (see also next sentence)* **We only defined the climatic precession in the figure caption. We clarify that in the text P11 L8-11 "On figure 12, we plotted the values of the climatic precession (defined as e*sin(w – 180°) with e the eccentricity and *w* the precession), which modulates the northern hemisphere insolation, and obliquity, which controls the temperature contrast between the high and low latitudes, relative to the SLPa-5 barycentre position, in order to analyse the relationship between the astronomical parameters and the SLPa-5 barycentre's position.+**

*p. 11, l. 3: the barycenter position changes with both longitude and latitude, please mention* **We change the sentence to "Climatic precession influences the SLPa-5 barycenter position both in latitude and longitude"**

*p. 11, l. 17: add 'marine sediment' before 'cores'* **Ok**

*p. 11, l. 23: 'limates'+* **Ok**

*p. 11, l. 30: which latitude range are you talking about? Shouldn't a high obliquity (= high summer insolation) reduce the latitudinal temperature gradient?* **We talk about the latitudes used to compute the ΔTT value (10°N-45°N) and (25°S-10°N). In our simulations a high obliquity, increases the summer insolation in the northern hemisphere high latitudes and reduces it in the low latitudes. This higher summer insolation in the NH warms the continent and increases the tropospheric temperature gradient in boreal summer, which is characteristic of a stronger summer monsoon.**

*p. 12, l. 8: replace 'solar' by 'northern hemisphere summer'* **Ok**

*p. 12, l. 21: 'imulations'+* **Ok**

*p. 12, l. 24: here is a contradiction to what you say later (l. 34). First you say that there is a mismatch to Bassinot et al., later you explain why your model matches with Bassinot et al., please clarify.*
**We mean that simulated productivity on the whole coastal area on which we made our analysis is not consistent with Bassinot et al. reconstructions. However, if we only focus on the northern part of the coastal area, our results match Bassinot et al's reconstruction for the mid-Holocene. Still they do not match the early Holocene reconstruction.**
**We have changed the formulation : "However, even if the simulated coastal summer productivity compares quite well to the data Rostek et al. 1997 (except for the LGM, their core is in the southern part of the coastal area), it shows some discrepancies when compared to the reconstructions in Bassinot et al. 2011 (their core is in the northern part of the coastal area)."**

*p. 13, l. 13: please check use of 'globally'* **We change it to largely**

*p. 13, l. 17-24: this part belongs to the method section* **Ok we move it to the Method section**

*p. 13, l. 27: replace 'are' by 'is'* **Ok**

*p. 13, l. 31: region up to 68_N? Please check* **Yes same typo as previously. We change it to 28N.**

*p. 14, l. 2: I don't think that the difference between productivity and export production in particular is a major problem for model and data comparison. It is rather that there are too many unknowns on what is finally best describing the signals contained in sediment records. As it stands, the statement seems to open a completely new aspect, which demands to be further elaborated. However, since it is not the main focus of the study I'*

*suggest leaving it out.* **We have chosen to keep it here but state more clearly that it has to be viewed as a perspective of this work.**

*p. 14, l. 11-17: it seems to me that in this paragraph you write about productivity, but it is actually the export, which is explained. Please check.* **We wanted to explain primary productivity but it is true that we miss some arguments and that this description better fits the export production. We change it to make it clear.**

*p. 15, l. 3: add 'or less' before 'nutrients'* **Ok**

*p. 15, l. 20: replace 'on' by 'to'* **Ok**
*Again, this sentence seems to give a hint to problems of model data comparison. I'd suggest adding a full paragraph on what this study contributes to a improved model-data comparison or leave it out.*
**We change the sentence to: Í This study allows us to draw attention to certain points that may affect the reconstruction of past climate and productivity and hence the comparison between model and data.Î**

*p. 15, l. 24: replace 'should' by 'is expected to'* **Ok**

**Figures:**
*Figure 2, caption: 68_N? Probably this should be 28_N?* **Yes**
*Figure 8: caption:replace 'Dash' by 'Dashed'* **Ok**
*Figure 14: 'circles+or 'color of the circles'* **Ok**

**NOTE:**

We realized that the computing of the barycenter was erroneous: in the submitted version, the barycenter was computed by taking the SLP anomalies relative to the regional mean (20W-150E,30S-60N) and not to the global mean as we intended. We corrected that. This correction made us change figures 5, 11 and 12. The barycenters move slightly north, especially in some glacial climates, but this does not change our conclusions. The new figures are added at the end of this document.

We added a few word in section 3.2.2 p11 L2-7:

**‰This latitudinal movement of the SLPa-5 barycentre with the monsoon strength seems also to be dependent on the glacial or interglacial state of the simulation. Indeed, in glacial simulations the SLPa-5 barycentre is north of the CTRL barycentre even if the monsoons are less intense (LGM, MIS4D and MIS4M), however the other glacial simulations with higher summer monsoon intensity (MIS3 and MIS4F) have their barycentre located north of these glacial simulations (Fig. 10). Similarly, in inter-glacial climates, the Holocene simulations have a barycentre north of the CTRL one (Fig. 10).Î**

And we change a sentence on p.12 l.17-19
**‰The changes in the ITCZ position highlighted by Fleitmann et al. (2007) are consistent with our results, especially with the changes in the position of the SLPa-5 barycentre in latitude given a similar background state: glacial or interglacial** (Fig.11).+
instead of ‰The changes in the ITCZ position highlighted by Fleitmann et al. (2007) are consistent with our results, especially with 10 the changes in the position of the SLPa-5 barycentre in latitude (Fig.11).+

**List of all relevant changes made in the manuscript:**

1. We removed all the discussion and figures related to the central Arabian Sea as suggested by the reviewers.
2. We added a more developed paragraph on the model biases in our discussion.
3. We modified some of the figures to add more physical variables (in the evaluation for instance) and we re-organized some of the figure's order (Fig.14 becomes Fig.4)
4. We modified the computation of the SLPa-5 barycenter, which modifies figure 5, 11 and 12.

[revised manuscript text omitted]